# Non-equilibrium Annealed Adjoint Sampler

**Jaemoo Choi**[1,*]**, Yongxin Chen**[1]**, Molei Tao**[1]**, Guan-Horng Liu**[2,*]

[1]Georgia Institute of Technology, [2]FAIR at Meta, [*]Core contributors
{jchoi843, yongchen, mtao}@gatech.edu, ghliu@meta.com

## Abstract

Recently, there has been significant progress in learning-based diffusion samplers, which aim to sample from a given unnormalized density. Many of these approaches formulate the sampling task as a stochastic optimal control (SOC) problem using a canonical uninformative reference process, which limits their ability to efficiently guide trajectories toward the target distribution. In this work, we propose the **Non-Equilibrium Annealed Adjoint Sampler (NAAS)**, a novel SOC-based diffusion framework that employs annealed reference dynamics as a non-stationary base SDE. This annealing structure provides a natural progression toward the target distribution and generates informative reference trajectories, thereby enhancing the stability and efficiency of learning the control. Owing to our SOC formulation, our framework can incorporate a variety of SOC solvers, thereby offering high flexibility in algorithmic design. As one instantiation, we employ a lean adjoint system inspired by adjoint matching, enabling efficient and scalable training. We demonstrate the effectiveness of NAAS across a range of tasks, including sampling from classical energy landscapes and molecular Boltzmann distributions.

## 1 Introduction

A fundamental task in probabilistic inference is to draw samples from a target distribution $\nu(x)$ given only its unnormalized density function, where the normalizing constant is intractable. This challenge arises across a wide range of scientific domains, including Bayesian inference (Box and Tiao, 2011), statistical physics (Binder et al., 1992), proteins (Jumper et al., 2021; Bose et al., 2024), and molecular chemistry (Weininger, 1988; Tuckerman, 2023; Midgley et al., 2023). Traditional approaches predominantly leverage Markov Chain Monte Carlo (MCMC) algorithms, which construct a Markov chain whose stationary distribution is the target distribution $\nu$ (Metropolis et al., 1953; Neal, 2001; Del Moral et al., 2006). However, these methods often suffer from slow mixing times, leading to high computational cost and limited scalability.

To overcome this limitation, recent research has focused on *Diffusion Neural Samplers*—a family of learned stochastic differential equation (SDE)–based samplers in which the dynamics are parameterized by neural networks (Zhang and Chen, 2022; Vargas et al., 2023; Chen et al., 2024; Albergo and Vanden-Eijnden, 2024; Phillips et al., 2024; Chen et al., 2024). Among diffusion neural samplers, many of the prior SOC-based approaches (Zhang and Chen, 2022; Vargas et al., 2023; Domingo-Enrich et al., 2025; Behjoo and Chertkov, 2025) aim to solve the stochastic optimal control (SOC) problem to construct unbiased diffusion samplers. However, these methods typically learn the controlled dynamics starting from a stationary or uninformative base reference, and therefore do not fully leverage annealed reference dynamics that progressively guide trajectories toward the target distribution.

Motivated by this limitation, we propose **Non-Equilibrium Annealed Adjoint Sampler** (**NAAS**), a new SOC-based diffusion framework that employs annealed reference dynamics as the base SDE. The annealing structure provides a natural progression toward the target distribution and generates

39th Conference on Neural Information Processing Systems (NeurIPS 2025).

Table 1: Compared to prior diffusion samplers, our **Non-equilibrium Annealed Adjoint Sampler (NAAS)** effectively guides the sampling process toward the target distribution by leveraging annealed reference dynamics, without using importance weighted sampling (IWS) during training.

| Method | Require IWS | Annealed Reference SDE |
|---|---|---|
| PIS (Zhang and Chen, 2022) DDS (Vargas et al., 2023) | No | ✗ |
| LV-PIS & LV-DDS (Richter and Berner, 2024) | No | ✗ |
| iDEM (Akhound-Sadegh et al., 2024) | yes | ✗ |
| AS (Havens et al., 2025) | No | ✗ |
| PDDS (Phillips et al., 2024) | Yes | ✓ |
| CRAFT (Matthews et al., 2022) SCLD (Chen et al., 2024) | Yes | ✓ |
| **NAAS** (Ours) | No | ✓ |

informative reference trajectories, which substantially improve the stability and efficiency of learning the control. As demonstrated empirically, this design leads to enhanced sample quality and target matching, especially when compared with methods relying on static references. To the best of our knowledge, NAAS is the first diffusion sampler that integrates the stochastic optimal control (SOC) perspective with reference annealed SDEs.

Because our method is formulated within the SOC framework, it can naturally integrate diverse solver choices (Nüsken and Richter, 2021; Domingo-Enrich et al., 2025), allowing flexible adaptation to different optimization settings. In contrast to existing annealed samplers (Albergo and Vanden-Eijnden, 2024; Phillips et al., 2024; Chen et al., 2024), which rely on importance-weighted sampling (IWS), NAAS supports solver choices that are independent of resampling-based estimators. Among these possible instantiations, we adopt Adjoint Matching (Domingo-Enrich et al., 2025)—a recently proposed, scalable, and practical solver that achieves low-variance gradient estimation and stable optimization in high-dimensional SOC problems.

As shown in Figure 1, our SOC formulation can be decomposed into two complementary SOC subproblems: *(i)* learning a prior distribution $\mu$ from the initial distribution $\delta_0$ with standard base reference, and *(ii)* transporting $\mu$ to the target $\nu$ using controlled annealed dynamics. Together, these components define a controlled diffusion process that realizes the transport $\delta_0 \to \mu \to \nu$, resulting in a sampler whose terminal distribution matches the target. In practice, we employ an alternating scheme to update the two control functions associated to these subproblems. We evaluate the effectiveness of NAAS on standard synthetic benchmarks and a challenging molecular generation task involving alanine dipeptide, demonstrating stable convergence and strong sample quality across settings.

Our main contributions are summarized as follows:

- We introduce a SOC-based unbiased sampler where the reference dynamics is governed by an annealed SDE. This formulation allows sampling to begin from a well-behaved initial process that gradually transitions toward the target distribution.

- We develop an efficient optimization algorithm based on *adjoint matching*, which provides a sample-efficient, and scalable approach to solving high-dimensional SOC problems.

- We assess our method on standard synthetic benchmarks and an alanine dipeptide molecular generation task. Our approach consistently outperforms existing baselines, demonstrating superior sample quality and diversity.

## 2 Preliminary

Throughout the paper let $\nu$ be the target distribution in space $\mathcal{X}$, and let $U_1 : \mathcal{X} \to \mathbb{R}$ be an energy function of the distribution $\nu$, i.e. $\nu(\cdot) \propto e^{-U_1(\cdot)}$.

**Stochastic Optimal Control (SOC)** The SOC problem studies an optimization problem subjected to the controlled SDE while minimizing a certain objective. We focus on one type of SOC problems formulated as (Kappen, 2005; Todorov, 2007)

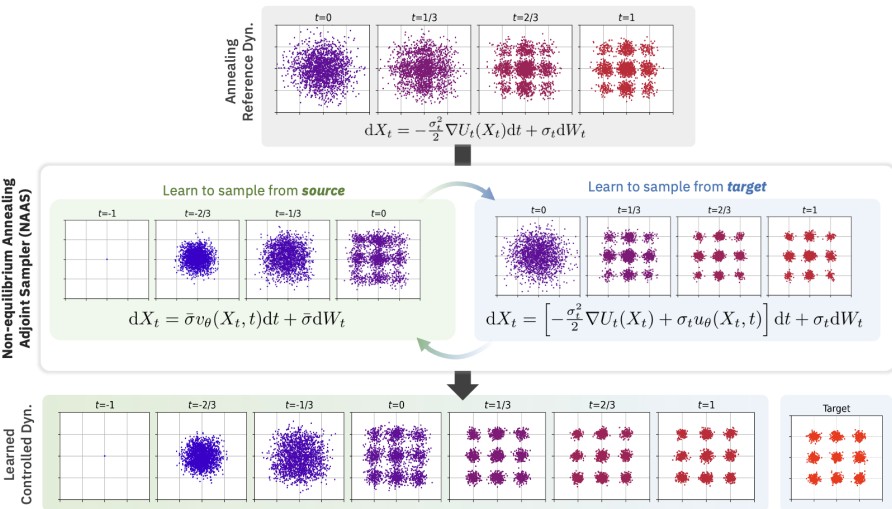

Figure 1: **Visualzation of Training Process of NAAS.** Given an annealing reference dynamics that provide high-quality—yet imperfect—initial samples in high-density regions of the target distribution $\nu$, NAAS learns to sample from both the target $\nu$ and the debiased source $\mu$ by alternate optimization between two control functions, $u^\theta$ and $v^\theta$, with Adjoint Matching (eq. (21)) and Reciprocal Adjoint Matching (eq. (24)). This iterative procedure progressively aligns the sampling path with the optimal control plan, leading to unbiased and efficient sampling from the target $\nu$.

$$\min_u \mathbb{E}\left[\int_0^1 \left(\frac{1}{2}\|u_t(X_t)\|^2 + f_t(X_t)\right)\mathrm{d}t + g(X_1)\right], \tag{1}$$

$$\text{s.t. } \mathrm{d}X_t = [b_t(X_t) + \sigma_t u_t(X_t)]\,\mathrm{d}t + \sigma_t\mathrm{d}W_t, \quad X_0 \sim \mu, \tag{2}$$

where $\mu$ is an initial distribution of random variable $X_0$, $b_t(X_t)$ is a given reference drift and $\sigma_t$ is the diffusion term. Here, we denote $f : [0,1] \times \mathcal{X} \to \mathbb{R}$ as the *running state cost* and $g : \mathcal{X} \to \mathbb{R}$ as the *terminal cost*. Throughout the paper, we denote $p^{\text{base}}$ and $p^\star$ as path measures induced by reference dynamics and optimal dynamics, respectively. The optimal control $u^\star$ is given by $u_t^\star(x) = -\sigma_t \nabla V_t(x)$, where the value function $V : [0,1] \times \mathcal{X} \to \mathbb{R}$ is defined as follows:

$$V_t(x) = \inf_u \mathbb{E}\left[\int_t^1 \left(\frac{1}{2}\|u_t(X_t)\|^2 + f_t(X_t)\right)\mathrm{d}t + g(X_1) \,\Bigg|\, X_t = x\right], \quad \text{s.t.} \quad eq. (2). \tag{3}$$

**Adjoint Matching (AM)** Naively backpropagating through the SOC objective (1) induces prohibitive computational costs. Instead, Domingo-Enrich et al. (2025) proposed a scalable and efficient approach, called Adjoint Matching (AM), to solving the SOC problem in (1). The AM objective is built on the insight that the optimal control policy $u^\star$ can be expressed in terms of the adjoint process. Specifically, AM constructs a surrogate objective by aligning the control function $u$ and a quantity derived from the adjoint process. The precise formulation is as follows:

$$\min_u \mathbb{E}_{X \sim p^{\bar{u}}}\left[\int_0^1 \|u_t(X_t) + \sigma_t\bar{a}(t;X)\|^2\mathrm{d}t\right], \quad \bar{a} = \text{stopgrad}(a), \quad \bar{u} = \text{stopgrad}(u), \tag{4}$$

$$\text{where} \quad \frac{\mathrm{d}}{\mathrm{d}t}a(t;X) = -\left[a(t;X)\nabla b_t(X_t) + \nabla f_t(X_t)\right], \quad a(1;X) = \nabla g(X_1). \tag{5}$$

Here, $X \sim p^{\bar{u}}$ denotes full trajectories sampled from the controlled dynamics using the current detached policy $\bar{u}$, and $\bar{a}$ represents a detached *lean* adjoint process $a(t;X)$, which is obtained by backward dynamics starting from the terminal condition $\nabla g(X_1)$.

**Reciprocal Adjoint Matching** Recently, Adjoint Sampling (AS) (Havens et al., 2025) propose a more efficient variant of adjoint matching tailored for the special case where both the drift term

and the running cost vanish, i.e. $b \equiv 0$ and $f \equiv 0$. In this setting, the lean adjoint $a(t; X)$ admits a closed-form expression; $a(t; X) = \nabla g(X_1)$, which is constant over time. Furthermore, they observe that since the optimal controlled path measure satisfies the *reciprocal property*:

$$p^\star(X) = p^\star(X_0, X_1)p^{\text{base}}(X|X_0, X_1), \tag{6}$$

the conditional law of intermediate states given the endpoints remains the same under both the optimal and the reference dynamics (Léonard et al., 2014). This insight simplifies the AM objective (4) to bypass the need to store full state-adjoint pairs $\{(X_t, a_t)\}_{t \in [0,1]}$. Instead, one can sample an intermediate state $X_t$ from the reference distribution $p^{\text{base}}_{t|0,1}(\cdot|X_0, X_1)$ given only the boundary states $(X_0, X_1)$. Note that since the drift term vanished, i.e. $b \equiv 0$, this conditional sampling is tractable.

**SOC-based (Memoryless) Diffusion Samplers**   We now consider the SOC problem (1) under the specific setting where

$$f \equiv 0, \quad g(x) := \log \frac{p^{\text{base}}_1(x)}{\nu(x)}, \quad p^{\text{base}}_{0,1}(X_0, X_1) = p^{\text{base}}_0(X_0)p^{\text{base}}_1(X_1). \tag{7}$$

The final condition implies that the initial and terminal states of the reference process are independent—a property we refer to as *memorylessness*. Under this assumption, the value function at the initial time, $V_0(x)$, becomes **constant** by Feynman-Kac formula (see Domingo-Enrich et al. (2025)). Then, applying Radon-Nikodym derivative on path measures $p^\star$ and $p^{\text{base}}$ yields the following (Dai Pra, 1991; Pavon and Wakolbinger, 1991; Chen et al., 2021)

$$\frac{\mathrm{d}p^\star_{0,1}(X_0, X_1)}{\mathrm{d}p^{\text{base}}_{0,1}(X_0, X_1)} \propto e^{-g(X_1)-V_0(X_0)} \Rightarrow p^\star_1(X_1) \propto \int_{\mathcal{X}} e^{-g(X_1)-V_0(X_0)} p^{\text{base}}_{0,1}(X_0, X_1)\mathrm{d}X_0,$$

$$\Rightarrow p^\star(X_1) \propto e^{-g(X_1)} p^{\text{base}}_1(X_1) = \nu(X_1).$$

That is, the terminal distribution induced by the optimal controlled SDE **exactly matches** the desired target $\nu$. Thus, solving the corresponding SOC problem leads to an unbiased diffusion-based sampler. Notably, several recent diffusion samplers can be understood within this memoryless SOC framework:

- **Path Integral Sampler** (**PIS**; Zhang and Chen, 2022) set $\mu = \delta_0$, where the reference dynamics automatically becomes memoryless. The reference dynamics is defined as follows:

$$\mathrm{d}X_t = \sigma \mathrm{d}W_t, \quad p^{\text{base}}_1(x) = \mathcal{N}(x; 0, \sigma^2 I). \tag{8}$$

  Several methods have been proposed to solve the associated SOC problem, including the adjoint system approach (Zhang and Chen, 2022), log-variance loss (LV) (Richter and Berner, 2024), and reciprocal adjoint matching (Havens et al., 2025).

- **Denoising Diffusion Sampler** (**DDS**; Vargas et al., 2023) This method constructs an approximately memoryless reference process using an Ornstein-Uhlenbeck (OU) dynamic. The initial distribution is set as $\mu = \mathcal{N}(0, \sigma^2 I)$, and the dynamics is given by:

$$\mathrm{d}X_t = -\frac{\beta_t}{2} X_t \mathrm{d}t + \sigma\sqrt{\beta_t}, \mathrm{d}W_t, \quad p^{\text{base}}_1(x) = \mathcal{N}(0, \sigma^2 I), \tag{9}$$

  where the annealing schedule $\beta_t$ is chosen so that $e^{\frac{1}{2}\int_0^1 \beta_t dt} \approx 0$, ensuring approximate independence between $X_0$ and $X_1$.

## 3   Non-Equilibrium Annealed Adjoint Sampler

In this section, we introduce **NAAS**, a novel SOC-based diffusion sampler that generates samples using non-equilibrium annealed processes. Specifically, NAAS is based on a newly designed SOC problem featuring annealed reference dynamics (Section 3.1). Since the problem requires sampling from an intractable prior $\mu$, we present a complementary SOC problem in Section 3.2 to approximate samples $X_0 \sim \mu$. Finally, Section 3.3 provides a detailed description of the algorithm and its practical implementation details. A detailed proof for theorems in this section are provided in Appendix A.

## 3.1 Stochastic Optimal Control with Annealing Paths

Let $U_0$ and $U_1$ be the energy potentials of some tractable prior (*e.g.*, Gaussian) and the desired target distribution $\nu$, respectively. We define a smooth interpolation between these potentials as follows:

$$U_t(x) := I(t, U_0(x), U_1(x)), \quad \text{with } U_{t=0}(x) = U_0(x), \quad U_{t=1}(x) = U_1(x), \tag{10}$$

where $I(t, \cdot, \cdot)$ denotes an interpolation scheme—such as linear or geometric interpolation. Based on this interpolation, we define an annealed SDE that serves as our reference dynamics:

$$\mathrm{d}X_t = -\frac{\sigma_t^2}{2}\nabla U_t(X_t)\mathrm{d}t + \sigma_t \mathrm{d}W_t, \quad X_0 \sim \mu. \tag{11}$$

We denote a path measure induced by the reference process as $p^{\text{base}}$. The following theorem introduces our newly designed SOC problem whose solution yields a sampler that transforms the specific prior distribution $\mu$ to the desired target distribution $\nu \propto e^{-U_1(\cdot)}$:

**Theorem 3.1** (SOC-based Non-equilibrium Sampler). *Let $\{U_t\}_{t \in [0,1]}$ be a time-dependent potential function defined in eq. (10). Consider the following SOC problem:*

$$\min_u \int_0^1 \mathbb{E}\left[\tfrac{1}{2}\|u_t(X_t)\|^2 + \partial_t U_t(X_t)\right]\mathrm{d}t,$$

$$\text{s.t. } \mathrm{d}X_t = \left[-\frac{\sigma_t^2}{2}\nabla U_t(X_t) + \sigma_t u_t(X_t)\right]\mathrm{d}t + \sigma_t \mathrm{d}W_t, \quad X_0 \sim \mu, \tag{12}$$

*where the initial distribution $\mu$ is defined as*

$$\mu(x) \propto \exp\left(-U_0(x) - V_0(x)\right), \tag{13}$$

*and $V_t(x)$ is the value function associated with the corresponding path measure. Note that $V_1(x) \equiv 0$ due to the absence of terminal cost. Then, the marginal distribution $p_t^\star(x)$ of the optimally controlled process admits the following form:*

$$p_t^\star(x) \propto \exp\left(-U_t(x) - V_t(x)\right), \quad \text{for all } t \in [0, 1]. \tag{14}$$

*In particular, the terminal distribution satisfies $p_1^\star(x) \propto \exp\left(-U_1(x)\right) =: \nu(x)$. Hence, the optimal controlled dynamics generates the desired target distribution $\nu$.*

**Sketch of Proof**  We outline the proof to provide an intuition behind the design of the objective function in the SOC formulation of eq. (12). Suppose $\varphi_t(x) = e^{-V_t(x)}$ and $\hat{\varphi}_t(x) = p_t^\star(x)e^{V_t(x)}$. Then, the pair $(\varphi, \hat{\varphi})$ satisfies the following PDEs, namely Hopf-Cole transform (Hopf, 1950):

$$\begin{cases} \partial_t \varphi_t(x) = \frac{\sigma_t^2}{2}\nabla U_t \cdot \nabla \varphi_t - \frac{\sigma_t^2}{2}\Delta\varphi_t + \partial_t U_t \varphi_t, & \varphi_0(x)\hat{\varphi}_0(x) = \mu(x), \\ \partial_t \hat{\varphi}_t(x) = \frac{\sigma_t^2}{2}\nabla \cdot (\nabla U_t \hat{\varphi} + \nabla\hat{\varphi}_t) - \partial_t U_t \hat{\varphi}_t, & \varphi_1(x)\hat{\varphi}_1(x) = \nu(x). \end{cases} \tag{15}$$

Note that the $\partial_t U_t$ terms in both PDEs arise from the running state cost in the SOC objective. The inclusion of this term allows us to derive closed-form expression of $\hat{\varphi}$:

$$\hat{\varphi}_t(x) = Ce^{-U_t(x)},$$

up to some multiplicative constant $C$. By the definition, we obtain

$$p_t^\star(x) = \varphi_t(x)\hat{\varphi}_t(x) \propto e^{-U_t(x)}\varphi_t(x) = \exp(-U_t(x) - V_t(x)),$$

which recovers eq. (14) claimed in the theorem.

**Remark.**  Importantly, the SOC problem (12) generally has an **intractable prior distribution** $\mu$, making the direct sampling from $\mu$ non-trivial. In Section 3.2, we propose a practical method to obtain a sample from $\mu$.

**Interpretation of Theorem 3.1**  In this paragraph, we highlight a special case in which $\mu$ becomes tractable. This case reveals an insightful connection to Denoising Diffusion Samplers (DDS) (Vargas et al., 2023). Since $U_0$ is prescribed and often tractable, $\mu \propto e^{-U_0(x)-V_0(x)}$ is tractable when $V_0(x)$ is constant. This can be achieved by setting the memoryless reference dynamics from $[0, \tau)$ where $U_{t \in [0,\tau)} := U_0$ for some $\tau < 1$, which ensures that the uncontrolled dynamics are sufficiently mixed between $t \in [0, \tau]$. In particular, if we choose $\tau = 1$, our SOC problem (12) reduces to DDS (Vargas et al., 2023):

**Corollary 3.2.** *When $U_{t \in [0,1)}(x) := U_0(x) = \frac{\|x\|^2}{2\bar{\sigma}^2}$ and $\beta_t = \bar{\sigma}^2 \sigma_t^2$, the problem* (12) *degenerates to*

$$\min_u \mathbb{E} \left[ \int_0^1 \frac{1}{2} \|u_t(X_t)\|^2 dt + U_1(X_1) - U_0(X_1) \right],$$

$$\text{s.t. } dX_t = \left[ -\frac{\beta_t}{2} X_t + \bar{\sigma} \sqrt{\beta_t} u_t(X_t) \right] dt + \bar{\sigma} \sqrt{\beta_t} dW_t, \quad X_0 \sim \mu, \tag{16}$$

*and setting $\mu \propto e^{-U_0}$ and $\beta_t$ to a memoryless noise schedule recovers DDS formulation in eq.* (9)*.*

Because $U_{t \in [0,1)}(x) = U_0(x)$, the annealing mechanism is absent in DDS, implying that the reference dynamics does not guide the particles toward the desired target distribution $\nu$.

## 3.2 Debiasing Annealed Adjoint Sampler through Learning Prior

From now on, we consider the specific case where the initial potential is given by $U_0(x) = \frac{\|x\|^2}{2\bar{\sigma}^2}$, i.e. $e^{-U_0} \propto \mathcal{N}(0, \bar{\sigma}^2 I)$. We begin by formulating a stochastic optimal control (SOC) problem whose optimal dynamics generates samples $X_0$ from the (intractable) prior distribution $\mu$.

**Lemma 3.3.** *Let $U_0(x) = \frac{\|x\|^2}{2\bar{\sigma}^2}$. Given the value function $V : \mathcal{X} \times [0,1] \to \mathbb{R}$ of the SOC problem eq.* (12)*, consider the following SOC problem:*

$$\min_v \mathbb{E} \left[ \int_{-1}^0 \frac{1}{2} \|v_t(X_t)\|^2 dt + V_0(X_0) \right],$$

$$\text{s.t. } dX_t = \bar{\sigma} v_t(X_t) dt + \bar{\sigma} dW_t, \quad X_{-1} \sim \delta_0. \tag{17}$$

*Note that the time interval of the underlying dynamics is $[-1, 0]$. Then, the terminal distribution induced by optimal control dynamics is $\mu := e^{-U_0(\cdot) - V_0(\cdot)}$.*

By the definition of the value function in eq. (3), $V_0(x)$ is the expected cost of the SOC problem (12) conditioned by $X_0 = x$. Thus, $V_0(\cdot)$ in eq. (17) can be replaced by eq. (12). As a result, we arrive at a unified SOC problem that defines an unbiased annealed sampling framework:

**Theorem 3.4.** *Consider the following SOC problem:*

$$\min_{u,v} \mathbb{E} \left[ \int_{-1}^0 \frac{1}{2} \|v_t(X_t)\|^2 dt + \int_0^1 \frac{1}{2} \|u_t(X_t)\|^2 dt + \int_0^1 \partial_t U_t(X_t) dt \right], \tag{18}$$

$$\text{s.t. } dX_t = \bar{\sigma} v_t(X_t) dt + \bar{\sigma} dW_t \quad for \ -1 \le t < 0, \quad X_{-1} \sim \delta_0, \tag{19}$$

$$dX_t = \left[ -\frac{\sigma_t^2}{2} \nabla U_t(X_t) + \sigma_t u_t(X_t) \right] dt + \sigma_t dW_t \quad for \quad 0 \le t \le 1. \tag{20}$$

*Then, the optimal control dynamics is an unbiased sampler, i.e., $p_1^\star = \nu \propto e^{-U_1}$.*

## 3.3 Practical Implementation

In this section, we present a practical algorithm for solving the SOC problem in eq. (18). Our approach **alternates** between updating the control policies $u$ and $v$ for the annealed dynamics (20) and the prior-estimation dynamics (19), respectively. We employ adjoint matching to gain a scalability and efficiency. The following two paragraphs describes the training objective for the controls $u$ and $v$.

**Adjoint Matching for Annealed Dynamics** We now describe how to optimize the control function $u_t(x)$ for annealed control dynamics eq. (20). Let $u^\theta$ be the parameterization of this control. Given an initial state $X_0$ sampled from eq. (19), we simulate the forward trajectory $X = \{X_t\}_{t \in [0,1]}$ using the current control estimate $u^\theta$. To update the control $u^\theta$, we adapt the AM framework eq. (4), yielding

$$\mathcal{L}(u^\theta; X) = \int_0^1 \|u_t^\theta(X_t) + \sigma_t a(t; X)\|^2 dt, \quad X \sim p^{\bar{u}}, \quad \bar{u} = \text{stopgrad}(u^\theta), \tag{21}$$

$$\text{where } \frac{da(t; X)}{dt} = -\left[ a(t; X) \cdot \left( -\frac{\sigma_t^2}{2} \nabla^2 U_t(X_t) \right) + \partial_t \nabla U_t(X_t) \right], \ a(1; X) = 0, \tag{22}$$

---

**Algorithm 1** Non-equilibrium Annealed Adjoint Sampler (NAAS)

---

**Require:** Tractable energy $U_0$, twice-differentiable target energy $U_1(x)$, two parametrized networks $u^\theta : [0,1] \times \mathcal{X} \to \mathcal{X}$ and $v^\theta : [-1,0] \times \mathcal{X} \to \mathcal{X}$ for control parametrization, buffers $\mathcal{B}_u$ and $\mathcal{B}_v$. Number of epochs $N_u$ and $N_v$.

1: **for** stage $k$ **in** $1, 2, \ldots$ **do**
2:     **for** epoch **in** $1, 2, \ldots N_u$ **do**
3:         Sample $\{X_t\}_{t \in [-1,1]}$ through eq. (19) and eq. (20).         ▷ forward pass
4:         Compute lean adjoint $\{a_t\}_{t \in [0,1]}$ through eq. (22).         ▷ backward pass
5:         Push $\{(t, a_t, X_t)\}_{t \in [0,1]}$ into a buffer $\mathcal{B}_u$.         ▷ add to buffer
6:         Optimize $u^\theta$ by eq. (21) with samples from $\mathcal{B}_u$.         ▷ adjoint matching
7:     **end for**
8:     **for** epoch **in** $1, 2, \ldots N_v$ **do**
9:         Sample $\{X_t\}_{t \in [-1,1]}$ through eq. (19) and eq. (20).         ▷ forward pass
10:        Compute lean adjoint $\{a_t\}_{t \in [0,1]}$ through eq. (22).         ▷ backward pass
11:        Push $(X_0, a_0)$ into a buffer $\mathcal{B}_v$.         ▷ add to buffer
12:        Optimize $v^\theta$ by eq. (24) and (25) with samples from $\mathcal{B}_v$.   ▷ reciprocal adjoint matching
13:     **end for**
14: **end for**

---

where $\nabla^2 U_t$ is the Hessian of $U_t(x)$ whose vector-Hessian product can be computed efficiently by

$$a(t; X) \cdot \left( -\frac{\sigma_t^2}{2} \nabla^2 U_t(X_t) \right) = -\frac{\sigma_t^2}{2} \nabla \left( \bar{a}(t; X) \nabla U_t(X_t) \right) \quad \bar{a} = \text{stopgrad}(a). \tag{23}$$

Note that this AM scheme is not as computationally efficient as AS (Havens et al., 2025), since it requires solving a backward ODE due to the use of annealed reference dynamics. Nevertheless, it remains more efficient than alternative approaches such as naive backpropagation or the log-variance method proposed in Richter and Berner (2024).

**Reciprocal Adjoint Matching for Sampling the Prior**   Similarly, we address the optimization of the control function $v_t(x)$ corresponding to eq. (19), parameterized as $v^\theta$. We employ the reciprocal AM framework to learn $v^\theta$. Specifically, we begin by sampling the initial states $X_0$ from controlled dynamics with current control $v^\theta$. We then generate full trajectories $X = \{X_t\}_{t \in [0,1]}$ using the annealed dynamics associated with $u^\theta$, and simulate the backward adjoint process via eq. (22). Then, we obtain a state-adjoint pair $(X_0, a_0)$ on $t = 0$. Since the reference dynamics w.r.t. $v$ has vanishing drift and state cost, we can apply reciprocal adjoint matching (Havens et al., 2025) as follows:

$$\mathcal{L}(v^\theta; X) = \int_{-1}^{0} \|v_t^\theta(X_t) + \sigma_t a_0\|^2 dt, \quad \bar{u} = \text{stopgrad}(u), \tag{24}$$

$$\text{where} \;\; X_t \sim p_{t|0}^{\text{base}}(\cdot | X_0) := \mathcal{N}\left( (1+t)X_0, (1+t)(2-t)\bar{\sigma}^2 I \right). \tag{25}$$

Thus, to update $v^\theta$, it is only required to cache state-adjoint pair $(X_0, a_0)$ at $t = 0$, which is memory efficient. For the detailed derivation of conditional distribution, see Appendix A.

**Replay Buffers**   Computing the full forward-backward system at every iteration is computationally expensive. To address this, we employ replay buffers that store the necessary state-adjoint pairs used to train the control networks. Specifically, instead of recomputing the lean adjoint system at each step, we refresh the buffer every 200–500 iterations, which significantly reduces computational overhead. For training $u^\theta$, we maintain a buffer $\mathcal{B}_u$ containing triplets $(t, X_t, a_t)$ for $t \in [0,1]$. For training $v^\theta$, we use a separate buffer $\mathcal{B}_v$ to store the state-adjoint pair $(X_0, a_0)$ at $t = 0$. The implementation details are provided in Appendix D.

**Algorithm**   As described in Algorithm 1, we optimize the control functions $u^\theta$ and $v^\theta$ in an alternating fashion using the adjoint matching objective desribed in eq. (21) and eq. (24). To compute the required state-adjoint pair, we solve the forward-backward systems defined in eq. (22) and eq. (25). To avoid the computational overhead of solving these systems at every iteration, we adopt a replay buffer strategy that caches previously computed state-adjoint pairs.

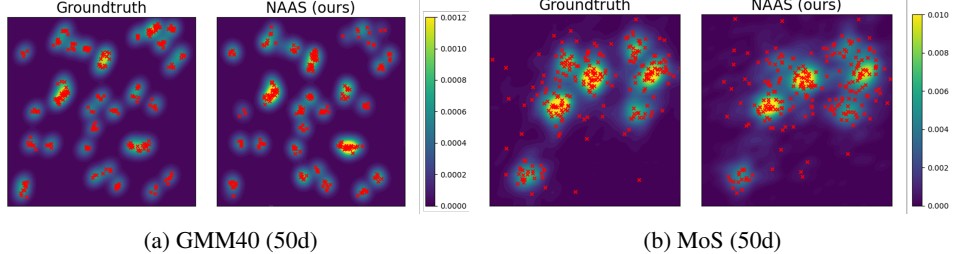

(a) GMM40 (50d)  (b) MoS (50d)

Figure 2: **Qualitative Comparison on GMM40 (50d) and MoS (50d).** For each task, we visualize the kernel density estimate (KDE) of the target distribution alongside the generated samples (shown as red dots), projected onto the first two principal axes. The samples from NAAS closely follow the structure and support of the ground-truth, demonstrating accurate mode coverage and high sample fidelity in high-dimensional settings.

**The role of two optimization stages**  The two optimization problems in our framework, *(i)* learning the prior distribution $\mu$ over the interval $[-1, 0]$, and *(ii)* optimizing the controlled annealed dynamics over $[0, 1]$, exhibit fundamentally different learning dynamics and objectives.

- The first-stage optimization over $[-1, 0]$ learns a prior distribution $\mu = e^{-U_0 - V_0}$, where $U_0$ is a simple, prescribed energy function (typically corresponding to Gaussian distribution). This prior is designed to have broad and smooth support, in contrast to the often sharp or multimodal structure of the target distribution $\nu$. The objective here is not to capture fine-scale structure, but to construct a low-complexity distribution that can be easily sampled using a simple base process, while still broadly covering the support of $\nu$.

- In contrast, the second-stage optimization over $[0, 1]$ focuses on refining the annealed dynamics to guide trajectories from $\mu$ toward the target distribution $\nu$. Since the annealed reference dynamics already bias the flow toward high-density regions of the target, the control at this stage primarily serves to correct residual mismatches. Because we only required to learn the residual mismatch, the second optimization problem is also easy to learn.

By decoupling these two objectives, the alternating scheme enables more stable and effective training. Each optimization stage solves a simpler and more focused subproblem: the first constructs a well-behaved initialization, and the second incrementally corrects the flow toward the target. This separation reduces optimization interference and improves both convergence and sample quality.

## 4  Related Works

**Sequential Monte Carlo and Annealed Sampler**  Many recent works have focused on Markov Chain Monte Carlo (MCMC) and Sequential Monte Carlo (SMC) methods (Chopin, 2002; Del Moral et al., 2006; Guo et al., 2024), which sequentially sample from a series of intermediate (often annealed) distributions that bridge a tractable prior and a complex target distribution. A large number of these methods leverage annealed importance sampling (AIS) (Neal, 2001; Guo et al., 2025) to correct for discrepancies between the proposal and target distributions, often improving the quality of proposals over time. Several earlier approaches that do not rely on neural networks (Wu et al., 2020; Geffner and Domke, 2021; Heng et al., 2020) have been developed to learn transition kernels for MCMC. Later, transition kernels has been explicitly parameterized using normalizing flows (Dinh et al., 2014; Rezende and Mohamed, 2015), enabling more expressive and tractable models. Based on this idea, various approaches (Arbel et al., 2021; Matthews et al., 2022) have been proposed by refining SMC proposals via variational inference with normalizing flows. More recently, diffusion-based annealed samplers (Vargas et al., 2024; Akhound-Sadegh et al., 2024; Phillips et al., 2024), inspired by score-based generative modeling (Song et al., 2021), have gained attention. These methods utilize annealed dynamics derived from denoising diffusion processes and typically combine importance sampling with a matching loss to account for the mismatch between the sampling trajectory and the target distribution. On other line of work, Sequential Control for Langevin Dynamics (SCLD) (Chen et al., 2024) improves training efficiency by employing off-policy optimization over annealed dynamics. However, it still relies on importance sampling to ensure accurate marginal sampling.

Table 2: Results on synthetic energy functions. We report Sinkhorn distance (↓) and MMD (↓). References for each comparison method are provided in Table 1 and Section 5. All comparison values are taken from SCLD (Chen et al., 2024).

| Method | MW54 ($d=5$) Sinkhorn | Funnel ($d=10$) MMD | Funnel ($d=10$) Sinkhorn | GMM40 ($d=50$) Sinkhorn | MoS ($d=50$) MMD | MoS ($d=50$) Sinkhorn |
|---|---|---|---|---|---|---|
| SMC | $20.71 \pm 5.33$ | - | $149.35 \pm 4.73$ | $46370.34 \pm 137.79$ | - | $3297.28 \pm 2184.54$ |
| SMC-ESS | $1.11 \pm 0.15$ | - | $\mathbf{117.48} \pm 9.70$ | $24240.68 \pm 50.52$ | - | $1477.04 \pm 133.80$ |
| CRAFT | $11.47 \pm 0.90$ | $0.115 \pm 0.003$ | $134.34 \pm 0.66$ | $28960.70 \pm 354.89$ | $0.257 \pm 0.024$ | $1918.14 \pm 108.22$ |
| DDS | $0.63 \pm 0.24$ | $0.172 \pm 0.031$ | $142.89 \pm 9.55$ | $5435.18 \pm 172.20$ | $0.131 \pm 0.001$ | $2154.88 \pm 3.861$ |
| PIS | $0.42 \pm 0.01$ | - | - | $10405.75 \pm 69.41$ | $0.218 \pm 0.007$ | $2113.17 \pm 31.17$ |
| AS | $0.32 \pm 0.06$ | - | - | $18984.21 \pm 62.12$ | $0.210 \pm 0.004$ | $2178.60 \pm 54.82$ |
| CMCD-KL | $0.57 \pm 0.05$ | $0.095 \pm 0.003$ | $513.33 \pm 192.4$ | $22132.28 \pm 595.18$ | - | $1848.89 \pm 532.56$ |
| CMCD-LV | $0.51 \pm 0.08$ | - | $139.07 \pm 9.35$ | $4258.57 \pm 737.15$ | - | $1945.71 \pm 48.79$ |
| SCLD | $0.44 \pm 0.06$ | - | $134.23 \pm 8.39$ | $3787.73 \pm 249.75$ | - | $656.10 \pm 88.97$ |
| **NAAS** (Ours) | $\mathbf{0.10} \pm 0.0075$ | $\mathbf{0.076} \pm 0.004$ | $132.30 \pm 5.87$ | $\mathbf{496.48} \pm 27.08$ | $\mathbf{0.113} \pm 0.070$ | $\mathbf{394.55} \pm 29.35$ |

## 5 Experiments

In this section, we evaluate NAAS across a diverse set of sampling benchmarks. Further implementation details are provided in Appendix D.

- Our primary focus on challenging synthetic distributions, including the 32-mode many-well (MW54, 5d) distribution, the 10d Funnel benchmark, a 40-mode Gaussian mixture model (GMM40) in 50d, and a 50d Student mixture model (MoS). We assess the performance using the maximum mean discrepancy (MMD) following Akhound-Sadegh et al. (2024) and the Sinkhorn distance (Sinkhorn) with a small entropic regularization ($10^{-3}$) following Chen et al. (2024).

- We demonstrate the practical applicability of NAAS by extending our evaluation to the generation of Alanine Dipeptide (AD) molecular structures. This molecule consists of 22 atoms in 3D. Following the implementation of Midgley et al. (2023); Wu et al. (2020), we use the OpenMM library (Eastman et al., 2017) and represent the molecular structure using internal coordinates, resulting in a 60-dimensional state space. We evaluate KL divergence ($D_{KL}$) between 2000 generated and reference samples for the backbone angles $\phi, \psi$ and the methyl rotation angles $\gamma_1, \gamma_2, \gamma_3$, which are referred to as *torsions*. Moreover, we report Wasserstein distance between energy values of generated and reference samples ($E(\cdot)W_2$).

**Main Results**   As shown in Table 2, our proposed method, NAAS, consistently outperforms existing baselines across the majority of synthetic sampling benchmarks. On the MW54 and GMM40 tasks, NAAS improves over the second-best method by more than 75%, demonstrating exceptionally strong performance. Furthermore, on both the Funnel (MMD) and MoS benchmarks, NAAS achieves substantial improvements over competing methods. These results collectively highlight the effectiveness of NAAS as a high-quality sampler across diverse and challenging targets.

Figure 2 presents qualitative results for the GMM40 and MoS benchmarks, showing side-by-side visualizations of samples generated by NAAS and the corresponding ground truth. The generated samples accurately capture the structure and support of the true distributions. All modes are well-represented without collapse or over-concentration, indicating high sample diversity and fidelity.

Figure 3 illustrates the training dynamics of NAAS, with both MMD and Sinkhorn distance plotted over training iterations. In particular, the MoS experiment Figures 3b and 3c highlight that our initial dynamics are remarkably strong; our model begins training with an MMD around 0.22 and a Sinkhorn distance near 2600. These values are already competitive with, or even superior to, the final performance achieved by several established baselines such as SMC, SMC-ESS (Buchholz et al., 2021), PIS (Zhang and Chen, 2022), and DDS (Vargas et al., 2023). All comparison benchmarks, including SMC, SMC-ESS, and CMCD (Vargas et al., 2024) are taken from SCLD (Chen et al., 2024). This favorable starting point is a direct consequence of the annealed reference dynamics employed by our method, which provides a meaningful initialization that captures coarse features of the target distribution. NAAS then further improves the sampler by learning the control. Ultimately, NAAS surpasses the second-best method, SCLD (Chen et al., 2024), by more than 30% in both metrics,

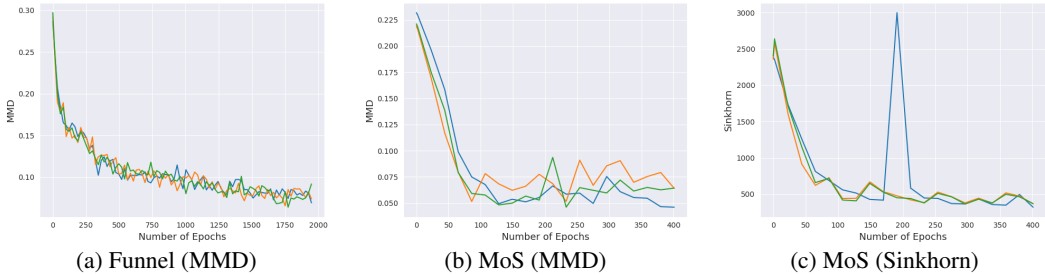

| (a) Funnel (MMD) | (b) MoS (MMD) | (c) MoS (Sinkhorn) |

Figure 3: Visualization of training dynamics over epochs. Each runs are presented in a different color.

demonstrating that its performance gains are not merely due to favorable initialization, but also to its ability to effectively learn corrections to the reference dynamics. This highlights the strength of NAAS in combining informative priors with principled learning for efficient and high-fidelity sampling.

**Alanine Dipeptide Generation** We compare our model to PIS (Zhang and Chen, 2022) and DDS (Vargas et al., 2023), which can be viewed as a special case of our framework that excludes the annealed path measure. As shown in Table 3, our model achieves significantly better performance in terms of $\gamma_1$ and $E(\cdot)W_2$, while showing comparable results on the remaining metrics. It also achieves favorable values across all torsion angles while maintaining a similar energy distribution, indicating the absence of mode collapse. Furthermore, as illustrated by the qualitative results in Figure 4, the torsion angle distributions produced by our model closely follow the trends of the reference density.

Table 3: Results for AD generation. We report KL divergence for torsions ($D_{\mathrm{KL}}$) and Wasserstein 2-distance for energies $E(\cdot)W_2$.

| Method | $\phi$ | $\psi$ | $\gamma_1$ | $\gamma_2$ | $\gamma_3$ | $E(\cdot)W_2$ |
|---|---|---|---|---|---|---|
| PIS | 0.597 | 0.952 | 0.453 | 0.498 | 7.038 | 5.918 |
| DDS | 0.493 | 0.154 | 4.095 | 0.111 | 0.150 | 5.467 |
| **NAAS** (Ours) | 0.260 | 0.236 | 0.272 | 0.132 | 0.156 | 1.076 |

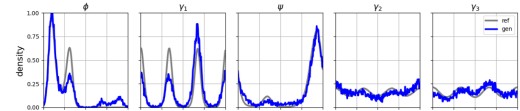

Figure 4: Comparison of five torsions between generated and reference samples

## 6  Conclusion

We proposed NAAS, a novel and unbiased sampling algorithm grounded in SOC theory under annealed reference dynamics. Our key contribution is on formulating the SOC problem that results unbiased diffusion sampler tailored to have annealed reference process. To construct an unbiased estimator, we propose a joint SOC problem involving two distinct control functions, which are optimized in an alternating manner using an adjoint matching. We validated NAAS on a diverse set of synthetic benchmarks, demonstrating state-of-the-art performance across all tasks. Additionally, we extended our framework to molecular structure generation with alanine dipeptide, suggesting that NAAS can scale to more complex and structured domains. Despite these encouraging results, it remains to be tested its effectiveness on larger-scale and real-world scenarios, potentially with more expressive annealing schedules. Moreover, a deeper investigation into the trade-off between the two optimization stages—examining how the strength of the annealing term influences the behavior of NAAS—remains an important direction for future work. As a limitation, NAAS requires computing the Hessian of the given energy function during sample generation, which introduces computational overhead. Moreover, NAAS requires two networks to train due to its alternating scheme. We do not see any negative societal consequences of our work that should be highlighted here.

## Acknowledgments and Disclosure of Funding

JC and YC acknowledge support from NSF Grants ECCS-1942523, DMS-2206576, and CMMI-2450378. MT is thankful for partial supports by NSF Grants DMS-1847802, DMS-2513699, DOE Grants NA0004261, SC0026274, and Richard Duke Fellowship.

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

# A  Proofs and Additional Theorems

## A.1  Proof of Theorem 3.1

*Proof.* Let $V : [0, 1] \times \mathcal{X} \to \mathbb{R}$ be a value function defined as eq. (3). Then, the necessary condition, namely Hamilton-Jacobi-Bellman (HJB) equation (Bellman, 1954), is written as follows:

$$0 = \partial_t V_t(x) - \frac{\sigma_t^2}{2} \nabla U_t(x) \cdot \nabla V_t(x) + \frac{\sigma_t^2}{2} \Delta V_t(x) - \frac{\sigma_t^2}{2} \|\nabla V_t(x)\|_2^2 + \partial_t U_t(x), \quad V_1(x) = 0, \tag{26}$$

where $u_t^\star(x) = -\sigma_t \nabla V_t(x)$. Moreover, the corresponding Fokker-Planck equation (FPE) (Risken, 1996) can be written as follows:

$$\partial_t p_t^\star(x) = -\nabla \cdot \left( \left( -\frac{\sigma_t^2}{2} \nabla U_t(x) - \sigma_t^2 \nabla V_t(x) \right) p_t^\star(x) \right) + \nabla^2 \cdot \left( \frac{\sigma_t^2}{2} p_t^\star(x) \right) \tag{27}$$

Now, let $\varphi_t(x) = e^{-V_t(x)}$ and $\hat{\varphi}_t(x) = p_t^\star(x) e^{V_t(x)}$. Then,

$$\partial_t \varphi_t(x) = -\partial_t V_t(x) e^{-V_t(x)},$$

$$= \left( -\frac{\sigma_t^2}{2} \nabla U_t(x) \cdot \nabla V_t(x) + \frac{\sigma_t^2}{2} \Delta V_t(x) - \frac{\sigma_t^2}{2} \|\nabla V_t(x)\|_2^2 + \partial_t U_t(x) \right) e^{-V_t(x)}, \tag{28}$$

$$= \frac{\sigma_t^2}{2} \nabla U_t(x) \cdot \nabla \varphi_t(x) - \frac{\sigma_t^2}{2} \Delta \varphi_t(x) + \partial_t U_t(x) \varphi_t(x).$$

Following the eq. (52) in Liu et al. (2022), we can derive

$$\partial_t \hat{\varphi}_t = \exp(V_t) \left( \frac{\partial p_t^\star}{\partial t} + p_t^\star \frac{\partial V_t}{\partial t} \right)$$

$$= \exp(V_t) \left( \nabla \left( \frac{\sigma_t^2}{2} \nabla U_t p_t^\star \right) + \sigma_t^2 \Delta V_t p_t^\star + \sigma_t^2 \nabla V_t \cdot \nabla p_t + \frac{\sigma_t^2}{2} \Delta p_t^\star \right)$$

$$+ \exp(V_t) p_t^\star \left( \frac{\sigma_t^2}{2} \nabla U_t \cdot \nabla V_t - \frac{\sigma_t^2}{2} \Delta V_t + \frac{\sigma_t^2}{2} \|\nabla V_t\|_2^2 - \partial_t U_t \right)$$

$$= \frac{\sigma_t^2}{2} \underbrace{\exp(V_t) \left( 2\Delta V_t p_t^\star + 2\nabla V_t \cdot \nabla p_t + \Delta p_t^\star + \|\nabla V_t\|_2^2 - \Delta V_t \right)}_{=\Delta \hat{\varphi}_t}$$

$$+ \frac{\sigma_t^2}{2} \underbrace{\exp(V_t) \left( \nabla \left( \nabla U_t p_t^\star \right) + p_t^\star \nabla V_t \nabla U_t \right)}_{\nabla \cdot (\nabla U_t \hat{\varphi}_t)} - \partial_t U_t \underbrace{\exp \left( V_t \right) p_t^\star}_{=: \hat{\varphi}_t}$$

$$= \frac{\sigma_t^2}{2} \Delta \hat{\varphi}_t + \frac{\sigma_t^2}{2} \nabla \cdot (\nabla U_t \hat{\varphi}_t) - \partial_t U_t \hat{\varphi}_t.$$

Therefore,

$$\partial_t \hat{\varphi}_t(x) = \frac{\sigma_t^2}{2} \nabla \cdot (\nabla U_t \hat{\varphi} + \nabla \hat{\varphi}_t) - \partial_t U_t \hat{\varphi}_t.$$

Therefore, by combining these equation, we obtain a joint PDE, namely a Hopf-Cole transform (Hopf, 1950), i.e.

$$\begin{cases} \partial_t \varphi_t(x) = \frac{\sigma_t^2}{2} \nabla U_t \cdot \nabla \varphi_t - \frac{\sigma_t^2}{2} \Delta \varphi_t + \partial_t U_t \varphi_t, & \varphi_0(x) \hat{\varphi}_0(x) = \mu(x), \\ \partial_t \hat{\varphi}_t(x) = \frac{\sigma_t^2}{2} \nabla \cdot (\nabla U_t \hat{\varphi} + \nabla \hat{\varphi}_t) - \partial_t U_t \hat{\varphi}_t, & \varphi_1(x) \hat{\varphi}_1(x) = \nu(x). \end{cases} \tag{29}$$

Notice that $p_t^\star(x) = \varphi_t(x) \hat{\varphi}_t(x)$ and $\varphi_1(x) = $ constant by the definition. Moreover, by simple calculation, one can derive that $\hat{\varphi}$ has the following closed-form expression:

$$\hat{\varphi}_t(x) = C e^{-U_t(x)},$$

up to some multiplicative constant $C$. Combining these equations we obtain

$$p_t^\star(x) = \varphi_t(x) \hat{\varphi}_t(x) \propto e^{-U_t(x)} \varphi_t(x) = \exp(-U_t(x) - V_t(x)).$$

Hence,

$$p_0^\star(x) \propto \exp(-U_0(x) - V_0(x)), \quad p_1^\star(x) \propto \exp(-U_1(x) - V_1(x)) = C \exp(-U_1(x)) \propto \nu(x). \tag{30}$$

$\square$

## A.2 Proof of Corollary 3.2

*Proof.* Let $U_{t\in[0,1)}(x) := U_0(x) = \frac{\|x\|^2}{2\bar{\sigma}^2}$ and $\beta_t = \sigma_t^2/\bar{\sigma}^2$. Then,

$$-\frac{\sigma_t^2}{2}\nabla U_t(x) = -\frac{\sigma_t^2}{2}\nabla U_0(x) = -\frac{\beta_t}{2}x.$$

Thus, by substituting all these variables into eq. (12), the equation degenerates to

$$\min_u \mathbb{E}\left[\int_0^1 \frac{1}{2}\|u_t(X_t)\|^2 dt + U_1(X_1) - U_0(X_1)\right], \tag{31}$$

$$\text{s.t. } dX_t = \left[-\frac{\beta_t}{2}X_t + \bar{\sigma}\sqrt{\beta_t}u_t(X_t)\right] dt + \bar{\sigma}\sqrt{\beta_t}dW_t, \quad X_0 \sim \mu.$$

By Feynman-Kac formula (Le Gall, 2016),

$$\exp\left(-V_0(X_0)\right) = \mathbb{E}_{X\sim p^{\text{base}}(X|X_0)}\left[\exp\left(-(U_1(X_1) - U_0(X_1))\right)\right], \tag{32}$$

$$\overset{eq.\ (7)}{=} \mathbb{E}_{X_1\sim p_1^{\text{base}}(\cdot)}\left[\exp\left(-(U_1(X_1) - U_0(X_1))\right)\right] = \text{constant}. \tag{33}$$

Therefore, by substituting $V_0 \equiv$ constant to eq. (30), we obtain

$$p_0^\star(x) = C\exp(-U_0) = \mathcal{N}(0, \bar{\sigma}^2 I), \tag{34}$$

for some constant $C$. $\qquad\square$

Note that since

$$p^{\text{base}}(X_1|X_0) = \mathcal{N}\left(X_1; e^{-\frac{1}{2}\int_0^1 \beta_t dt}X_0, \bar{\sigma}^2\left(1 - e^{-\int_0^1 \beta_t dt}\right)I\right), \tag{35}$$

the reference dynamics is (almost) memoryless when $\int_0^1 \beta_t dt \approx \infty$.

## A.3 Proof of Lemma 3.3

*Proof.* Since $\hat{\varphi}_0(x) \propto e^{-U_0(x)}$,

$$V_0(X_0) = \log \hat{\varphi}_0(X_0) - \log \mu(X_0) = \log \mathcal{N}(x; 0, \hat{\sigma}^2 I) - \log \mu(X_0) + \text{constant},$$

the SOC problem (17) can be rewritten as follows:

$$\min_u \mathbb{E}\left[\int_{-1}^0 \frac{1}{2}\|u(X_t, t)\|^2 dt + \log \frac{\mathcal{N}(X_0; 0, \bar{\sigma}^2 I)}{\mu(X_0)}\right], \tag{36}$$

$$\text{s.t.} \quad dX_t = \bar{\sigma}u(X_t, t)dt + \bar{\sigma}dW_t, \quad X_{-1} \sim \delta_0.$$

By setting the terminal target distribution as $\mu$, this optimization problem becomes identical to the formulation in eq. (8), which we refer to as PIS. Consequently, the terminal distribution induced by its optimal solution is guaranteed to match $\mu$. $\qquad\square$

## A.4 Proof of Theorem 3.4

*Proof.* Let $V : [-1, 1] \times \mathcal{X} \to \mathbb{R}$ be a value function defined on the given SOC problem (18). Then, the corresponding HJB equation can be written as follows:

$$0 = \partial_t V_t(x) + \frac{\bar{\sigma}^2}{2}\Delta V_t(x) - \frac{\bar{\sigma}^2}{2}\|\nabla V_t(x)\|^2 \quad \text{for } t < 0, \quad eq.\ (26) \quad \text{for } t \geq 0. \tag{37}$$

Let $\varphi_t(x) = e^{-V_t(x)}$ and $\hat{\varphi}_t(x) = p_t^\star(x)e^{V_t(x)}$ for $-1 \leq t \leq 1$. Then, similar to the proof of Theorem 3.1, one can easily derive the following paired PDE:

$$\begin{cases} \partial_t \varphi(x, t) = -\frac{\bar{\sigma}^2}{2}\Delta\varphi \\ \partial_t \hat{\varphi}(x, t) = \frac{\bar{\sigma}^2}{2}\Delta\hat{\varphi} \end{cases} \quad \text{when } t < 0, \quad \varphi(x, 1) = 1, \tag{38}$$

$$\begin{cases} \partial_t \varphi(x, t) = \frac{\sigma_t^2}{2}\nabla U_t \cdot \nabla\varphi - \frac{\sigma_t^2}{2}\Delta\varphi + \partial_t U_t\varphi \\ \partial_t \hat{\varphi}(x, t) = \frac{\sigma_t^2}{2}\nabla \cdot (\nabla U_t\hat{\varphi} + \nabla\hat{\varphi}) - \partial_t U_t\hat{\varphi} \end{cases} \quad \text{when } t \geq 0, \tag{39}$$

As shown in the proof of Theorem 3.1, one can check that the solution (for some multiplicative constant $C$) to eq. (39) is

$$\hat{\varphi}_t(x) = Ce^{-U_t(x)}, \quad t \geq 0. \tag{40}$$

Therefore, $\hat{\varphi}_0(x) = Ce^{-U_0(x)}$ for some multiplicative constant $C$.

Now, over the interval $t \in [-1, 0]$, the function $\hat{\varphi}$ evolves according to the heat equation. Thus, we can explicitly write the solution for $\hat{\varphi}$ on $[-1, 0]$ given the terminal condition eq. (40) as follows:

$$\hat{\varphi}_t(x) = C'e^{-\frac{\|x\|^2}{2\bar{\sigma}^2(t+1)}} \quad \text{for } -1 < t < 0,$$

for some multiplicative constant $C'$. Note that $\lim_{t \to -1} \hat{\varphi}_t \propto \delta_0$. Combining these equation, $\hat{\varphi}$ admits a explicit solution, which can be written in a piecewise form over the full domain:

$$\hat{\varphi}_t(x) = \begin{cases} C'e^{-\frac{\|x\|^2}{2\bar{\sigma}^2(t+1)}} & \text{for } -1 < t < 0, \\ Ce^{-U_t(x)} & \text{for } 0 \leq t \leq 1. \end{cases} \tag{41}$$

Thanks to the explicitly written $\hat{\varphi}_t$, we can easily derive the following conditions:

$$\varphi_{-1}\hat{\varphi}_{-1} = \delta_0, \quad \varphi_0\hat{\varphi}_0 \propto e^{-V_0 - U_0} \propto \mu, \quad \varphi_1\hat{\varphi}_1 \propto e^{-U_1} \propto \nu. \tag{42}$$

Therefore, the solution to our stochastic optimal control (SOC) formulation in (18) yields the desired unbiased sampler. $\square$

### A.5 Detailed Explanation of Reciprocal Adjoint Matching

By using the lean adjoint matching loss, we can update both $u^\theta$, $v^\theta$. Precisely, the original lean adjoint matching loss can be written as follows:

$$\mathcal{L}(u^\theta, v^\theta; X) = \int_{-1}^0 \|v_t^\theta(X_t) + \bar{\sigma}a(t; X)\|^2 dt + \int_0^1 \|u_t^\theta(X_t) + \sigma_t a(t; X)\|^2 dt, \tag{43}$$

$$\text{where} \quad X \sim p^{(\bar{v}, \bar{u})}, \quad (\bar{v}, \bar{u}) = \text{stopgrad}((v^\theta, u^\theta)), \quad a(1; X) = 0, \tag{44}$$

$$\frac{da(t; X)}{dt} = -\left[a(t; X) \cdot \left(-\frac{\sigma_t^2}{2}\nabla^2 U_t(X_t)\right) + \partial_t \nabla U_t(X_t)\right], \quad t \geq 0, \tag{45}$$

$$\frac{da(t; X)}{dt} = 0, \quad t < 0. \tag{46}$$

Thus, in the backward interval $t < 0$, the lean adjoint variable remains constant: $a(t; X) = a(0; X) =: a_0$ for all $t < 0$. Furthermore, as discussed in eq. (6), the optimal backward control $v^\theta$ satisfies the reciprocal property. That is, under the optimal dynamics, the marginal distribution of $X_t$ given the endpoints $X_{-1} = 0$ and $X_0$ corresponds to a conditional Gaussian:

$$X_t \sim p_{t|-1,0}^{\text{base}}(\cdot \mid X_{-1} := 0, X_0) \Rightarrow X_t \sim \mathcal{N}\left((t+1)X_0, (t+1)(2-t)\bar{\sigma}^2 I\right), \tag{47}$$

for all $t \in [-1, 0)$. By leveraging this optimality condition, we obtain our reciprocal adjoint matching objective as eq. (24).

## B Further Discussion on NAAS

### B.1 Discussion on Scalability and Algorithmic Stability

This section elaborates on three aspects of NAAS: (i) scalability with two learnable control functions, (ii) the rationale behind the alternating optimization scheme, and (iii) the complexity and stability of Algorithm 1.

**Scalability** Although NAAS employs two control networks, the overall computational and memory costs remain comparable to existing neural control frameworks. Both control functions share the same lightweight architecture and are trained over disjoint annealing intervals, such that the doubled parameter count does not lead to doubled runtime or memory overhead. Empirically, both stages converge within similar iteration counts, and the per-iteration training time remains nearly unchanged since gradients are backpropagated through independent adjoint integrations. This modular design preserves scalability even in high-dimensional settings.

**Alternating Optimization**   The optimization of NAAS is deliberately decoupled into two stages, each targeting a distinct subproblem:

- **Stage 1** ($t \in [-1, 0]$) learns a smooth, broad prior distribution that provides wide support over the target space. This stage functions as a coarse-scale initialization similar to multiscale optimization methods.

- **Stage 2** ($t \in [0, 1]$) refines the control to transport samples from the coarse prior to the target distribution. As the annealed dynamics naturally bias trajectories toward high-density regions, this refinement primarily corrects residual mismatches, ensuring stability and data efficiency.

This alternating design mitigates interference between global and local objectives, leading to more stable convergence than a fully joint optimization. It also facilitates clearer diagnostics, as each stage's role—broad coverage versus fine refinement—can be independently verified.

**Empirical Comparison with Joint Optimization**   To verify the effect of our alternating algorithm, we compared it with a joint optimization scheme that trains a single control over the entire time horizon $t \in [-1, 1]$. Evaluations were conducted on the MW54 and MoS benchmarks. As shown in Table 4, our alternating optimization consistently outperforms the joint optimization across both tasks. These empirical findings align well with our theoretical motivation, demonstrating that decomposing the learning process into two focused stages yields better convergence and overall performance.

Table 4: Comparison between joint and alternating optimization schemes on MW54 and MoS. We report Sinkhorn distance ($\downarrow$).

| Scheme | MW54 | MoS |
|---|---|---|
| Joint | 0.20 | 597.99 |
| Alternating (Ours) | **0.10** | **394.55** |

## B.2   Difference between NAAS and NETS

**Non-equilibrium Sampling**   In a similar spirit to AIS (Neal, 2001), non-equilibrium samplers are methods that progressively guide samples over a finite horizon using a meaningful reference annealing path defined by the annealing energy $U_t$ between target and prior distributions, for instance, the same reference path eq. (11) is considered for both NAAS and NETS (Albergo and Vanden-Eijnden, 2024). Such guidance from non-equilibrium—or reference—distribution $\exp(-U_t)$ indeed plays a key role in these methods.

**Difference between NETS and NAAS**   NETS (Albergo and Vanden-Eijnden, 2024) apply Jarzynski's equality (Jarzynski, 1997) to match intermediate distribution at time $t$ to be proportional to $\exp(-U_t)$. So, in NETS, an additional component $u_t$ in drift term is introduced to eliminate the discrepancy between reference dynamics and the desired distribution $p_t \propto \exp(-U_t)$.

On the other hand, while NAAS also appends an additional term $u_t$ to the same reference path eq. (11), , its $u_t$ is derived within the stochastic optimal control framework, aiming to solve a specific objective formulated in eq. (12). This distinction alters the effect of $u_t$ in modifying the reference path eq. (11) between the two methods, as the objective of NAAS eq. (12) does not explicitly enforce $u_t$ to match the reference distribution $\exp(-U_t)$. Indeed, we prove in Section 3.1 that the proposed by NAAS results in a controlled SDE whose instantaneous distribution $p_t$ satisfies $p_t \propto \exp(-U_t - V_t)$, where $V_1 = 0$. That is, the instantaneous distribution of NAAS is highly correlated with the reference distribution $\exp(-U_t)$, but is biased by an additional term $\exp(-V_t)$, which vanishes at the terminal time $t = 1$. Furthermore, we highlight that since $V_1 = 0$, $p_1 \propto \exp(-U_1 - V_1) = \exp(-U_1) \propto \nu$. Hence, the controlled SDEs proposed by NAAS do converge to the desired target distribution $\nu$ at terminal time $t = 1$.

### B.3 Connection between NAAS and PDDS

**Particle Denoising Diffusion Sampler (PDDS)** (Phillips et al., 2024) considers a stochastic process between Normal distribution $\mathcal{N}$ and the target $\pi = \frac{\gamma}{Z}$

$$dY_t = [-\beta_{1-t}Y_t + 2\beta_{1-t}\nabla \log g_{1-t}(Y_t)]\, dt + \sqrt{2\beta_{1-t}}dB_t, \quad Y_0 \sim \mathcal{N}(0, I), \quad Y_1 \sim \pi = \frac{\gamma}{Z}, \tag{48}$$

where the "guidance" $g_t$ is given by

$$g_t(y) := \int g_0(x)p_0(x|x_t = y)dx, \qquad g_0 = \frac{\gamma}{\mathcal{N}}, \tag{49}$$

$$p(x_0|x_t) = \mathcal{N}(x_0; \sqrt{1 - \lambda_t}x_t, \lambda_t I), \qquad \lambda_t = 1 - \exp\left(-2 \int_0^t \beta_\tau d\tau\right) \tag{50}$$

and can be learned, $v(x, t; \theta) \approx \nabla \log g_{1-t}(x)$, via the NSM loss:[1]

$$\int_0^1 \mathbb{E}\|v(x, t; \theta) - \kappa_{1-t}\nabla \log g_0(X_0)\|^2 dt, \qquad \kappa_t := \sqrt{1 - \lambda_t} = \exp\left(-\int_0^t \beta_\tau d\tau\right), \tag{51}$$

where the expectation is taken over the optimal distribution corrected via IWS.

**SOC formulation for PDDS** In this paragraph, we introduce the SOC problem that corresponds to the optimal solution of PDDS. Moreover, we further discuss that the *adjoint matching loss* for the SOC problem is equivalent to NSM loss.

**Proposition B.1.** *Equation* (48) *is the optimal controlled process to the following SOC problem*

$$\min_u \mathbb{E}\left[\int_0^1 \tfrac{1}{2}\|u_t(X_t)\|^2 dt + \log \tfrac{\mathcal{N}}{\gamma}(X_1)\right], \tag{52}$$

$$\text{s.t. } dX_t = \left[-\beta_{1-t}X_t + \sqrt{2\beta_{1-t}}u_t(X_t)\right]dt + \sqrt{2\beta_{1-t}}dB_t, \quad X_0 \sim \mathcal{N}(0, I). \tag{53}$$

*That is, the optimal control to* (52) *is*

$$u_t^\star(x) = \sqrt{2\beta_{1-t}}\nabla \log g_{1-t}(x). \tag{54}$$

*Furthermore, the lean adjoint matching loss (w.r.t. the optimal distribution) for* (52) *is the same as the NSM loss* (51).

*Proof.* The optimal control to (52) follows

$$\begin{aligned}
u_t^\star(x) &= -\sqrt{2\beta_{1-t}}\nabla_x V_t(x)\\
&= -\sqrt{2\beta_{1-t}}\nabla_x \left(-\log \mathbb{E}_{X_1 \sim p_1(\cdot|X_t = x)}\left[\exp\left(-\log \tfrac{\mathcal{N}}{\gamma}(X_1)\right)\right]\right)\\
&= \sqrt{2\beta_{1-t}}\nabla_x \left(\log \int \tfrac{\gamma}{\mathcal{N}}(x_1)p_1(x_1|x_t = x)dx_1\right)\\
&= \sqrt{2\beta_{1-t}}\nabla \log g_{1-t}(x)
\end{aligned}$$

where the last equality follows by the fact that $p_t(x|x_s) = \mathcal{N}(x; \sqrt{1 - \lambda_{|t-s|}}x_s, \lambda_{|t-s|}I)$, due to $p_t = \mathcal{N}(0, I)$ for all $t \in [0, 1]$, and hence

$$\int \tfrac{\gamma}{\mathcal{N}}(y)p_1(y|x_t = x)dy = \int g_0(y)p_{1-2t}(y|x_{1-t} = x)dy = g_{1-t}(x).$$

Next, notice that the lean adjoint ODE in this case reads

$$\frac{d\tilde{a}_t}{dt} = \beta_{1-t}\tilde{a}_t, \qquad \tilde{a}_1 = \nabla \log \tfrac{\mathcal{N}}{\gamma}(X_1). \tag{55}$$

---

[1]We substitute the identity, $\nabla \log \pi_s(x) + x = \nabla \log g_s(x)$, into the original NSM loss, $\mathbb{E}\|\nabla \log \pi_s^\theta(X_t) + X_t - \kappa_s \nabla \log g_0(X_0)\|^2$.

The solution to (55) admits the form $\tilde{a}_t = C \exp\left(\int_0^t \beta_{1-s} ds\right)$, where $C$ is such that $\tilde{a}_1 = C \exp\left(\int_0^1 \beta_{1-s} ds\right) = \nabla \log \frac{\mathcal{N}}{\gamma}$. Hence, (55) can be solved in closed form

$$\tilde{a}_t = \nabla \log \tfrac{\mathcal{N}}{\gamma} \cdot \exp\left(\int_0^t \beta_{1-s} ds - \int_0^1 \beta_{1-s} ds\right)$$

$$= \nabla \log \tfrac{\mathcal{N}}{\gamma} \cdot \exp\left(-\int_t^1 \beta_{1-s} ds\right)$$

$$= \nabla \log \tfrac{\mathcal{N}}{\gamma} \cdot \exp\left(\int_{1-t}^0 \beta_\tau d\tau\right)$$

$$= -\nabla \log g_0 \cdot \kappa_{1-t},$$

where the third equality is due to change of variable $\tau := 1 - s$. Applying lean adjoint matching,

$$\mathbb{E}\|u(x,t;\theta) + \sqrt{2\beta_{1-t}}\tilde{a}_t\|^2 = \mathbb{E}\|u(x,t;\theta) - \sqrt{2\beta_{1-t}} \cdot \kappa_{1-t} \cdot \nabla \log g_0\|^2 \tag{56}$$

with a specific parametrization $u(x,t;\theta) = \sqrt{2\beta_{1-t}} \cdot v(x,t;\theta)$ yields the NSM loss (51). $\qquad\square$

**Connection to NAAS** The stochastic optimal control (SOC) problem underlying PDDS (Phillips et al., 2024)—formulated in eq. (52)—shares the same structure as the SOC problem defined in DDS (Vargas et al., 2023), i.e. eq. (9). As discussed in Section 3.1, the DDS objective is itself a special case of the more general SOC formulation introduced by NAAS (12). Therefore, the SOC perspective of PDDS naturally inherits this structure and can also be regarded as a special case within the broader NAAS framework. Despite the close connection in problem formulation, the methods used to solve the SOC problem are different. NAAS and DDS solve the SOC problem using adjoint-based learning objectives. In contrast, PDDS employs a non-adjoint-based objective, NSM loss (51). Instead, PDDS use the IWS strategy.

## B.4 Solving NAAS via Importance Sampling

In this section, we explore an alternative algorithmic approach for optimizing the SOC problem defined in eq. (18). While we do not empirically evaluate this method in the current work, we present it as a promising direction for future research. Our SOC formulation eq. (18) can be addressed in various method, resulting flexible algorithm design. Here, we provide another algorithm that handles our SOC problem based on important weighted sampling (IWS). Then, we introduce the close connection to PDDS (Phillips et al., 2024).

**Basic formula** Consider the SOC problem in eq. (1). In the seminar papers (Dai Pra, 1991; Pavon and Wakolbinger, 1991), the Radon-Nikodym (RN) derivative between the optimal and reference path measures can be written as follows:

$$\frac{dp^\star}{dp^{\text{base}}}(X) = \exp\left(-\int_0^1 f_t(X_t)dt - g(X_1) + V_0(X_0)\right), \tag{57}$$

Now, let $p^u$ be the path measure induced by the control $u$. By the Girsanov theorem (see Nüsken and Richter (2021)), we can derive

$$\frac{dp^\star}{dp^u}(X) = \frac{dp^{\text{base}}}{dp^u}(X)\frac{dp^\star}{dp^{\text{base}}}(X)$$

$$\propto \exp\left(\int_0^1 -\frac{1}{2}\|u_t(X_t)\|^2 dt - u_t(X_t) \cdot dW_t\right) \cdot \exp\left(-\int_0^1 f_t(X_t)dt - g(X_1)\right) / \varphi_0(X_0). \tag{58}$$

where

$$\varphi_0(X_0) = e^{-V_0(X_0)} = \mathbb{E}_{\mathbf{X}\sim p^{\text{base}}(\mathbf{X}|X_0)}\left[\exp\left(-\int_0^1 f_t(X_t)dt - g(X_1)\right)\right],$$

by Feynman-Kac formula (Le Gall, 2016).

**Importance Sampling for NAAS** Now, consider our SOC problem (18). Because $X_{-1} = 0$, eq. (58) can be rewritten as follows:

$$\frac{dp^\star}{dp^u}(X) \propto \exp\left(\int_{-1}^{0} -\frac{1}{2}\|v_t(X_t)\|^2 dt - v_t(X_t) \cdot dW_t\right)$$

$$\cdot \exp\left(\int_0^1 -\frac{1}{2}\|u_t(X_t)\|^2 dt - u_t(X_t) \cdot dW_t - \int_0^1 \partial_t U(X_t, t) dt\right). \quad (59)$$

Note that we can discard $\varphi_{-1}(X_{-1})$ since $X_{-1} = 0$. Given this expression, we can approximate the optimal path distribution $p^\star$ by sampling trajectories from $p^u$ and reweighting them using the Radon–Nikodym derivative (59). Specifically, the following importance sampling procedure allows us to recover samples from $p^\star$:

1. For given $N$, obtain a collection of trajectories $\{X^{(i)}\}_{i=1}^N$ where $X^{(i)} \sim p^u$.

2. Compute the importance weights $\{w^{(i)}\}_{i=1}^N$ where $w^{(i)} := \frac{dp^\star}{dp^u}(X^{(i)})$ by eq. (59).

3. Resample $\{X^{(i)}\}_{i=1}^N$ by following categorical distribution:

$$\hat{X} \sim \text{Cat}\left(\{\hat{w}^{(i)}\}_{i=1}^N, \{X^{(i)}\}_{i=1}^N\right), \quad (60)$$

where $\hat{w}^{(i)} = \frac{w^{(i)}}{\sum_{i=1}^N w^{(i)}}$.

**Importance Sampled Bridge Matching for NAAS** In this paragraph, we propose an IWS-based method for learning the optimal control $v^\star$ over the interval $t \in [-1, 0]$. Since $v^\star$ governs a transition from the Dirac distribution $\delta_0$ to the intermediate distribution $\mu$, this subproblem can be framed as a Schrödinger bridge problem (SBP) between $\delta_0$ and $\mu$. Specifically, $v^\star$ is the solution to the following stochastic control problem:

$$\min_v \mathbb{E}\left[\int_{-1}^0 \frac{1}{2}\|v_t(X_t)\|^2 dt\right] \quad \text{s.t.} \quad dX_t = \bar{\sigma} v_t(X_t) dt + \bar{\sigma} dW_t, \quad X_{-1} \sim \delta_0, \ X_0 \sim \mu. \quad (61)$$

Recent works (Shi et al., 2023; Liu et al., 2024) introduce the *bridge matching* (BM) loss as a scalable surrogate objective for solving SBPs. Under this framework, the optimal control $v^\star$ is obtained by minimizing following regression problem:

$$v^\star = \arg\inf_v \mathbb{E}_{X_0 \sim \mu}\left[\|v_t(X_t) - \bar{\sigma}\nabla\log p_{t|-1,0}^{\text{base}}(X_t|X_{-1}, X_0)\|^2\right]. \quad (62)$$

The only missing component is the ability to sample from the intractable intermediate distribution $\mu$. Given a current estimate of the control $v^\theta$, we can (asymptotically) sample from $\mu$ using importance-weighted sampling (IWS) as described in eq. (60). By combining IWS with the BM loss, we arrive at an iterative IWS-based algorithm for learning $v^\star$, as detailed in Algorithm 2.

## C  Additional Results

**Effect of Learning the Prior Distribution $\mu$** In this section, we demonstrate the effect of learning the prior distribution $\mu$ in eq. (24). To isolate the impact of the reciprocal update, we compare against a baseline where $v^\theta \equiv 0$ is held fixed. In other words, we remove the update of $v^\theta$ described in lines 8–14 of Algorithm 1. Theoretically, this corresponds to optimizing the SOC problem in eq. (12) under the fixed prior $\mu \propto e^{-U_0}$, which we refer to as a *biased* estimator. We conduct experiments on a simple two-dimensional Gaussian mixture model (GMM) target distribution, denoted *GMM-grid*, which the target distribution $\nu$ is defined as follows:

$$\nu(x) = \frac{1}{9}\sum_{i=1}^3\sum_{j=1}^3 \mathcal{N}(5(i-2, j-2), 0.3I). \quad (63)$$

As shown in Figure 5, both Sinkhorn and MMD measurement of the biased estimator quickly plateaus over training epochs, while the full NAAS algorithm continues to improve, achieving a sixfold improvement in performance. Moreover, as illustrated in Figure 6, NAAS provides significantly more uniform mode coverage compared to the biased variant, consistent with our theoretical result.

**Algorithm 2** IWS-NAAS

---

**Require:** Tractable energy $U_0$, twice-differentiable target energy $U_1(x)$, two parametrized networks $u^\theta : [0,1] \times \mathcal{X} \to \mathcal{X}$ and $v^\theta : [-1,0] \times \mathcal{X} \to \mathcal{X}$ for control parametrization, buffers $\mathcal{B}_u$ and $\mathcal{B}_v$. Number of epochs $N_u$ and $N_v$.

1: **for** stage $k$ **in** $1, 2, \ldots$ **do**
2:     **for** epoch **in** $1, 2, \ldots N_u$ **do**
3:         Sample $\{X_t\}_{t \in [-1,1]}$ through eq. (19) and eq. (20).      ▷ forward pass
4:         Compute lean adjoint $\{a_t\}_{t \in [0,1]}$ through eq. (22).      ▷ backward pass
5:         Push $\{(t, a_t, X_t)\}_{t \in [0,1]}$ into a buffer $\mathcal{B}_u$.      ▷ add to buffer
6:         Optimize $u^\theta$ by eq. (21) with samples from $\mathcal{B}_u$.      ▷ adjoint matching
7:     **end for**
8:     **for** epoch **in** $1, 2, \ldots N_v$ **do**
9:         Sample trajectories $\{X^{(i)}\}_{i=1}^N$ through eq. (19) and eq. (20).      ▷ forward pass
10:       Obtain resampled samples $\{\hat{X}_0^{(i)}\}_{i=1}^N$ through eq. (60).      ▷ IWS
11:       Push $\{\hat{X}_0^{(i)}\}_{i=1}^N$ into a buffer $\mathcal{B}_v$.      ▷ add to buffer
12:       Optimize $v^\theta$ by eq. (62) with samples from $\mathcal{B}_v$.      ▷ bridge matching
13:     **end for**
14: **end for**

---

**Effect of the Annealed Reference Dynamics** We investigate the role of annealed reference dynamics by varying the strength of the annealed reference dynamics. In the annealed SDE defined in eq. (11), stronger guidance can be achieved by increasing the diffusion scale $\sigma_t t \in [0,1]$. Following the *geometric noise schedule* (Song et al., 2021; Karras et al., 2022), we define $\sigma_t$ as

$$\sigma_t = \sigma_{\min}^t \sigma_{\max}^{1-t} \sqrt{2 \log \frac{\sigma_{\max}}{\sigma_{\min}}}, \tag{64}$$

where $\sigma_{\min}$ and $\sigma_{\max}$ are hyperparameters selected by design. Increasing $\sigma_{\max}$ effectively imposes a stronger guiding signal throughout the training. To assess its impact, we test three values of $\sigma_{\max}$—1000, 100, and 10—on the MoS (50d) distribution. As shown in Figure 7, we refer to these settings as *strong*, *moderate*, and *weak* guidance, respectively. As shown in the figure, stronger guidance improves training dynamics: larger $\sigma_{\max}$ results in lower initial Sinkhorn distance, faster convergence, and superior final performance. These results highlight the critical role of the annealed reference in steering the sampler during optimization.

**Results on High-dimensional Sampling Problems** We evaluate the scalability of NAAS on a high-dimensional 100D GMM40 task, comparing it with two strong baselines: AS (Havens et al., 2025), which employs standard reference dynamics without annealing, and SCLD (Chen et al., 2024), a state-of-the-art importance sampling method. We also report wall-clock time for the 50D setting, measured on a single NVIDIA A6000 GPU. As shown in Table 5, NAAS achieves superior sample quality and convergence stability, while maintaining competitive runtime. Its efficiency partly stems from the use of replay buffers, which stabilize optimization and enable sample reuse. Although solving the backward adjoint state eq. (22) introduces additional computational overhead, the resulting improvements in stability and sample quality justify this minor cost, highlighting the practicality and scalability of our approach.

Table 5: Results on the GMM40 benchmark across various dimensions.

| Method | $d = 4$ | $d = 16$ | $d = 50$ | $d = 100$ | Running time |
|---|---|---|---|---|---|
| SCLD (Chen et al., 2024) | 157.90 | 1033.19 | 3787.73 | 12357.49 | 30-40 mins |
| AS (Havens et al., 2025) | - | - | 18984.21 | - | 30 mins |
| NAAS (Ours) | **25.15** | **37.96** | **496.48** | **633.20** | 120 mins |

**Additional Experiments on Variance Measurement** Formally, IWS methods approximate the target distribution by $\nu \approx \sum_{i=1}^N w^{(i)} \delta_{X^{(i)}}$, where the unnormalized weights are obtained by $w^{(i)} \propto$

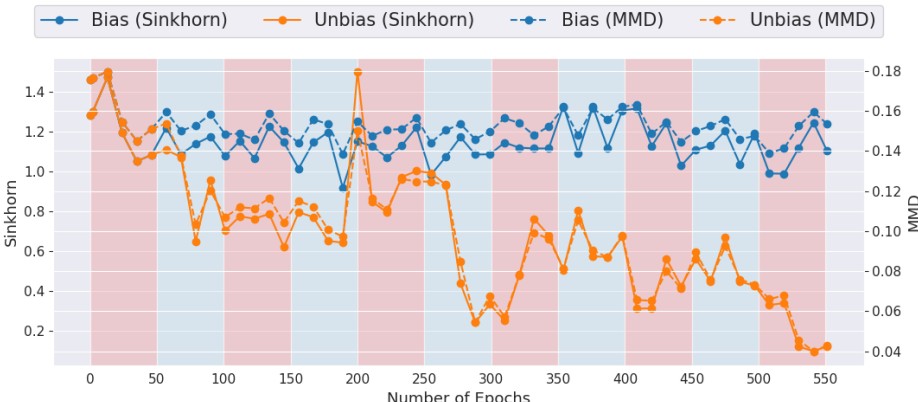

Figure 5: Training dynamics for the GMM-grid energy function over training epochs. We compare NAAS trained without prior correction (Biased, *blue*) and original NAAS (Unbias, *orange*). In the original NAAS setup, we update the control $v^\theta$ and $u^\theta$ in an alternating fashion every 50 epochs. The shaded regions in *lightblue* and *lightcoral* indicate periods during which $v^\theta$ and $u^\theta$ are updated, respectively. Our results show that the original NAAS consistently achieves lower Sinkhorn and MMD scores (both ↓), indicating that the prior correction term plays a key role in debiasing the sampler and improving sample quality.

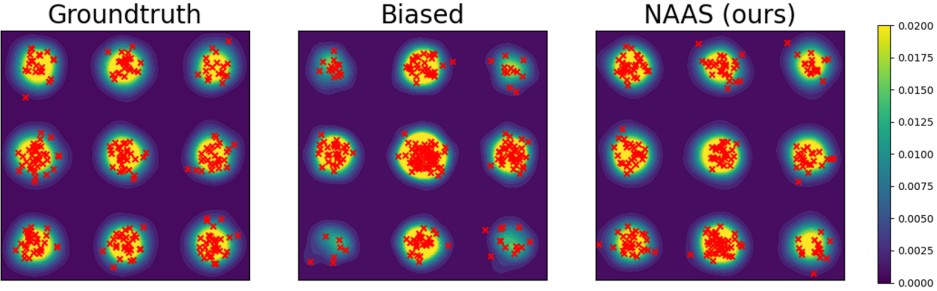

Figure 6: Side-by-side comparison of ground truth (*left*), NAAS w/o prior correction (Biased, *middle*), and NAAS (*right*). We visualize both the generated samples (red dots) and the corresponding kernel density estimates (KDE). As shown, the KDE plot for NAAS (Ours) exhibits more uniform coverage across all modes compared to the NAAS w/o correction, demonstrating that learning the prior distribution correctly debiases the sampler.

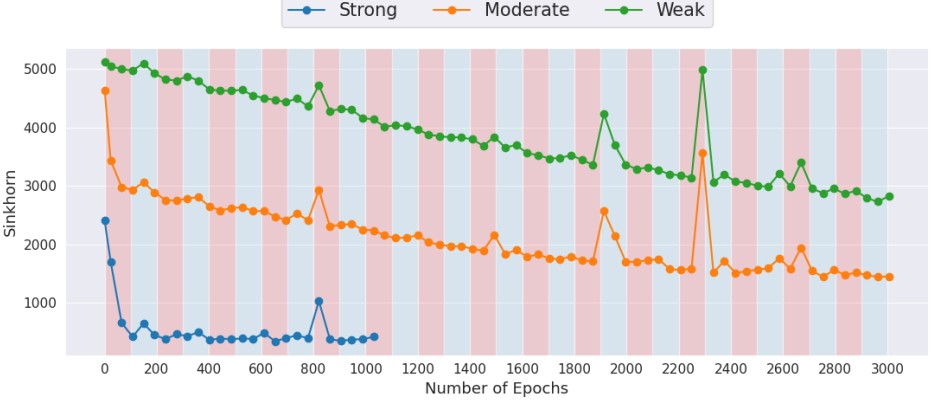

Figure 7: Visualization of Sinkhorn metric over training epochs for varying levels of reference guidance. We compare three variants of the annealed reference dynamics, each using a different guidance strength. Specifically, we set $\sigma_{\max} = 1000, 100$, and $10$ to represent *strong*, *moderate*, and *weak* guidance, respectively. As shown, stronger guidance leads to better initial performance, faster convergence, and superior final performance.

$\frac{\mathrm{d}p^\star}{\mathrm{d}p}(X^{(i)})$ with the ratio between the target path measure $p^\star$ and proposal $p$. When the support or mass of the proposal and target distributions are misaligned—as is common in high-dimensional or complex energy landscapes—these weights can suffer from high variance, as noted in prior works (Del Moral et al., 2006; Stoltz et al., 2010). This leads to poor effective sample size and significant variance in the empirical approximation, and can be further exacerbated when the number of samples $N$ is insufficient.

To see it in empirically, we report the variance of the importance weights across various benchmarks. We choose LV-PIS as the baseline for comparison as the method directly optimizes the variance of importance weights and, similar to our NAAS, starts the generation from Dirac delta. For NAAS, we compute its importance weights using eq. (59). As shown in Table 6, the variance of the importance weights for NAAS is significantly smaller than that of LV-PIS. Remarkably, NAAS achieves this reduction in variance despite never explicitly optimizing the variance of importance weights.

Table 6: Comparison on the Variance of Importance Weights ($\downarrow$)

| Method | MW54 | Funnel |
|---|---|---|
| LV-PIS (Richter and Berner, 2024) | 21.11 | 5.46 |
| NAAS (Ours) | **1.84** | **0.11** |

**Effect of Annealed Reference Dynamics** We conduct additional experiments to verify whether the improved performance of NAAS over AS arises from the use of annealing reference. Specifically, we instantiate NAAS with a particular choice of $U_t$, which remains equal to $U_0$ for all $t \in [0,1)$ and jump to $U_t = U_1$ at terminal $t = 1$. This setup effectively leads to a two-staged AS baseline that retains the same first process $[-1,0]$ as NAAS, while completely removing the annealing reference in the second stage. As we proved in Corollary 3.2, the optimal process of this two-staged AS baseline satisfies the desired target. As shown in Table 7, the two-stage AS (first row) performs similarly to the original AS. In contrast, our NAAS (second row) outperforms this baseline by a significant margin. These results highlight the important role of the annealed reference in NAAS, which contributes non-trivially to its improved performance.

Table 7: Comparison between two-staged AS (Havens et al., 2025) and NAAS. We report Sinkhorn distances ($\downarrow$).

| Method | Interpolation | MW54 | GMM40 |
|---|---|---|---|
| two-staged AS | $U_{t \in [0,1)} = U_0$ | 0.43 | 19776.91 |
| NAAS | $U_t = (1-t)U_0 + tU_1$ | **0.10** | **496.48** |

**Effect of $\sigma_t$** The diffusion term $\sigma_t$ on time $t \in [0,1]$ is a key hyperparameter of our method. This controls the guidance scale of annealed reference dynamics, so if $\sigma_t$ is large, then gives more guidance, small then small guidance. As shown in Table 8, we ablate the performance of NAAS on GMM40 benchmark by gradually increasing the $\sigma_{\max}$ that defines $\sigma_t := \sigma_{\min}^t \sigma_{\max}^{1-t} \sqrt{2 \log \frac{\sigma_{\max}}{\sigma_{\min}}}$.

As shown in Table 8, it is evident that the performance of NAAS depends on $\sigma_{\max}$, as the hyperparameter directly affects its exploration capability. However, once $\sigma_{\max}$ surpasses some threshold ( 20 for this specific benchmark), the performance of NAAS stabilizes and becomes insensitive to further increase of $\sigma_{\max}$. That said, tuning $\sigma_t$ in practice could be straightforwardly performed with e.g., grid or binary search, without having access to samples from or near target distribution. We emphasize these results as strong evidence of the robustness of NAAS for practical usages.

Table 8: Ablation studies on $\sigma_{\max}$. We report Sinkhorn distances ($\downarrow$).

| Method | $\sigma_{\max} = 1$ | $\sigma_{\max} = 10$ | $\sigma_{\max} = 20$ | $\sigma_{\max} = 50$ | $\sigma_{\max} = 100$ | $\sigma_{\max} = 200$ |
|---|---|---|---|---|---|---|
| NAAS | 22893.72 | 1350.76 | 639.86 | 496.48 | 443.55 | 492.83 |

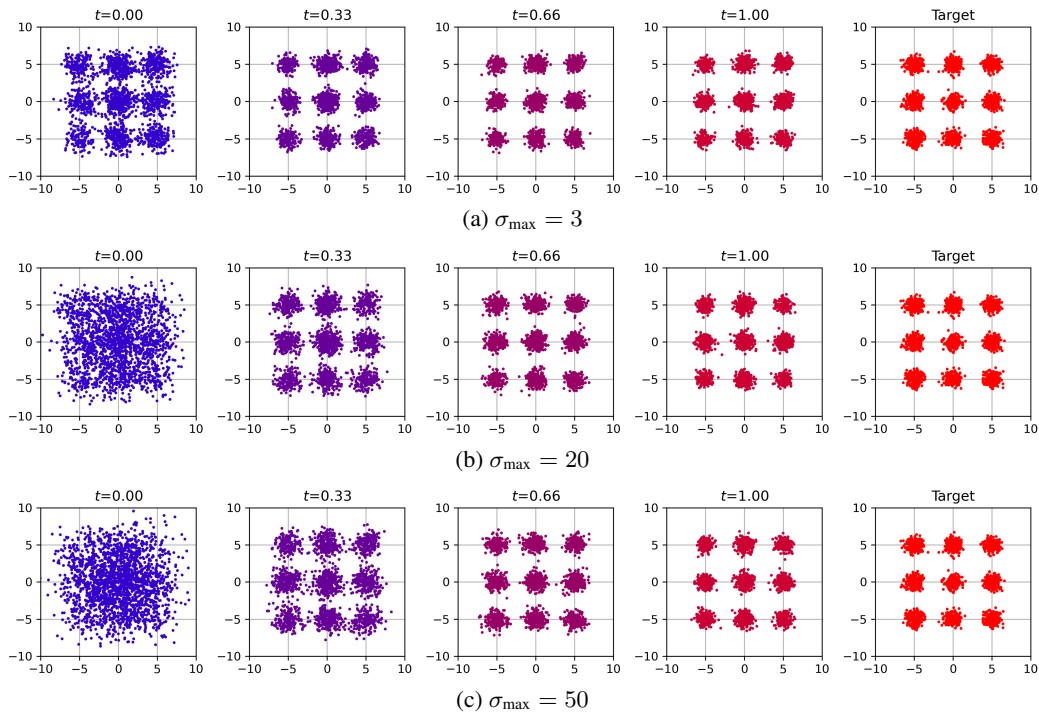

Figure 8: Ablation on $\sigma_{\max}$. As $\sigma_t$ increases, the prior distribution $\mu$ at $t = 0$ becomes increasingly similar to the stationary distribution $\propto e^{-U_0}$.

**Visualizing Samples for Various $\sigma_{\max}$**    We visualize the effect of varying the guidance strength, parameterized by $\sigma_t$. As shown in eq. (13), increasing $\sigma_t$ drives the prior distribution $\mu$ at $t = 0$ closer to the reference distribution, i.e., $\propto e^{-U_0}$. As illustrated in Figure 8, larger values of $\sigma_t$ make $\mu$ increasingly resemble a Gaussian distribution $\mathcal{N}(0, 3^2 I) \propto e^{-U_0}$, which is consistent with our theoretical expectation.

**Mode Coverage**    We further conduct experiments to assess whether the proposed sampler can accurately recover mode weights in multimodal distributions with non-uniform component weights. Following the setup introduced in LRDS (Noble et al., 2025), we consider a bimodal Gaussian mixture with components located at $\mathcal{N}(-a\mathbf{1}_d, \Sigma_1)$ and $\mathcal{N}(a\mathbf{1}_d, \Sigma_1)$, associated with mixture weights $w_1 = \frac{2}{3}$ and $w_2 = \frac{1}{3}$, respectively.

As reported in Table 9, our method achieves competitive performance and accurately recovers the true mode proportions, indicating that NAAS does not exhibit mode blindness or mode switching in this setting. Interestingly, we observe that the controlled dynamics in the first stage ($t \in [-1, 0]$) play a crucial role in reweighting the modes. At the beginning of training, the estimated weights are approximately $\hat{w}_1 \approx 0.5$, but as the control for $t \in [-1, 0]$ is learned, the sampler gradually reallocates mass toward $\hat{w}_1 \approx \frac{2}{3}$, aligning with the true distribution.

Table 10 summarizes our results across varying between-mode distances $a$. NAAS consistently recovers the correct mode weights and performs comparably to LRDS, even as the separation between modes increases. For LRDS, we retained the default hyperparameters but trained longer for $a = 1, 2, 3$ to ensure convergence; we note that its performance could likely be further improved with additional tuning. We emphasize, however, that these results are not intended as an extensive comparison between NAAS and LRDS, but rather as an ablation study demonstrating the robustness of NAAS in challenging multimodal scenarios with widely separated modes.

We also evaluate NAAS and LRDS on simpler benchmarks, MW54 and Funnel. As shown in Table 11, NAAS achieves slightly better overall performance than LRDS under these settings.

Table 9: Comparison on mode coverage. We report the absolute deviation in mode weight, $|\hat{w}_1 - w_1|$ ($\downarrow$).

| Method | $d = 16$ | $d = 32$ |
|--------|----------|----------|
| AS | 20.5 % | - |
| LRDS | 1.7 % | 2.7 % |
| NAAS | 3.3 % | 2.0 % |

Table 10: Ablation study on mode distance $a$ with dimension $d = 16$. We report the absolute deviation in mode weight, $|\hat{w}_1 - w_1|$ ($\downarrow$).

| Method | $a = 1$ | $a = 2$ | $a = 3$ |
|--------|---------|---------|---------|
| LRDS | 1.7 % | 6.5 % | 6.7 % |
| NAAS | 1.7 % | 3.2 % | 3.6 % |

Table 11: Comparison on MW54 and Funnel. We report Sinkhorn distance ($\downarrow$).

| Method | MW54 | Funnel |
|--------|------|--------|
| LRDS | 0.85 | 152.09 |
| NAAS | 0.10 | 132.30 |

# D  Implementation Details

In this section, we provide the detailed setup for our synthetic energy experiments in Table 1. We evaluate our model on four synthetic energy functions, namely MW-54, Funnel, GMM40, MoS following SCLD (Chen et al., 2024). These examples are typical and widely used examples to examine the quality of the sampler. Throughout the discussion, let $d$ be a dimension of the state and let

$$x = (x_1, x_2, \ldots, x_d).$$

**MW-54**  We use the same many-well energy function as in SCLD (Chen et al., 2024). We consider many-well energy function, where the corresponding unnormalized density is defined as follows:

$$\nu(x) \propto \exp - \sum_{i=1}^{m}(x_i^2 - \delta)^2 - \frac{1}{2}\sum_{m+1}^{d} x_i^2, \tag{65}$$

where $d = 5$, $m = 5$, and $\delta = 4$, leading to $2^m = 2^5 = 32$ wells in total.

**Funnel**  We follow the implementation introduced in Neal (2003); Chen et al. (2024), which is a funnel-shaped distribution with $d = 10$. Precisely, it is defined as follows:

$$\nu(x) \propto \mathcal{N}(x_1; 0, \sigma^2)\mathcal{N}((x_2, \ldots, x_d); 0, \exp(x_1)I), \tag{66}$$

with $\sigma^2 = 9$.

**GMM40**  Let the target distribution $\nu$ is defined as follows:

$$\nu(x) = \frac{1}{m}\sum_{i=1}^{m} \nu_i(x). \tag{67}$$

Following (Blessing et al., 2024; Chen et al., 2024), each mode of Gaussian Mixture Model (GMM) is defined as follows:

$$\nu_i = \mathcal{N}(m_i, I), \quad m_i \sim U_d(-40, 40), \tag{68}$$

where $U_d(a, b)$ is a uniform distribution on $[a, b]^d$. We use $d = 50$.

**MoS**  We also use the same Mixture of Student's t-distribution (MoS) following Blessing et al. (2024); Chen et al. (2024). Following the notation of eq. (67), we define MoS of $m = 10$ and $d = 50$, where each mode $\nu_i$ is defined as follows:

$$\nu_i = t_2 + m_i, \quad m_i \sim U_d(-10, 10), \tag{69}$$

where $t_2$ is a Student's t-distribution with a degree of freedom of 2.

**Alanine Dipeptide (AD)**  is a molecule that consists of 22 atoms in 3D, therefore, in total 66-dimension data. Following prior work (Zhang and Chen, 2022; Wu et al., 2020), we aim to sample from its Boltzmann distribution at temperature 300K. Moreover, we convert energy function to a internal coordinate with the dimension $d = 60$ through the OpenMM library Eastman et al. (2017).

Table 12: Hyperparameter Settings

| Hyperparameter | MW54 | Funnel | GMM40 | MoS | AD |
|---|---|---|---|---|---|
| $K$ | 3 | 10 | 5 | 5 | 10 |
| $(N_u, N_v)$ | (100, 100) | (100, 100) | (100, 100) | (100, 100) | (1000, 200) |
| $(M_u, M_v)$ | (400, 400) | (400, 400) | (400, 400) | (400, 400) | (2000, 2000) |
| $B$ | 512 | 512 | 512 | 512 | 512 |
| $|\mathcal{B}|$ | 10000 | 10000 | 10000 | 10000 | 10000 |
| $N$ | 2048 | 512 | 512 | 512 | 200 |
| $E_{\max}$ | 100 | 1000 | 1000 | 1000 | 1000 |
| $A_{\max}$ | - | 100 | 100 | 100 | 10 |
| $(\mathrm{lr}_u, \mathrm{lr}_v)$ | $(10^{-5}, 10^{-8})$ | $(10^{-4}, 10^{-4})$ | $(10^{-6}, 10^{-8})$ | $(10^{-4}, 10^{-6})$ | $(10^{-4}, 10^{-8})$ |
| $\bar{\sigma}$ | 1.0 | 1.0 | 50 | 15 | 1 |
| $\sigma_{\max}$ | 1.0 | 9 | 50 | 1000 | 2 |
| $\sigma_{\min}$ | 0.01 | 0.01 | 0.01 | 0.01 | 0.01 |
| gradient clip | 1.0 | 1.0 | 1.0 | 1.0 | 1.0 |

**Training Hyperparameters** We adopt the notation of the hyperparameter of Algorithm 1. For additional hyperparameters, let $B$ be the batch size, $\mathrm{lr}_u$ and $\mathrm{lr}_v$ be the learning rate for $u^\theta$ and $v^\theta$, respectively. Moreover, note that we use Adam optimizer (Kingma and Ba, 2014) with $(\beta_1, \beta_2) = (0, 0.9)$ for all experiments. Let $|\mathcal{B}| := |\mathcal{B}_u| = |\mathcal{B}_v|$ be the buffer size for both buffers, $K$ be a total number of stages (see line 1), and let $M_u$ and $M_v$ be a total number of iterations to optimize lines 6 and 12, respectively. Let $N$ be the number of samples generated in lines 3 and 9, hence, a total of $N$ samples are eventually added to the buffer in line 6 and 12. Moreover, for stability, we clip the maximum gradient of energy function $\nabla U_t$ with clip parameter of $E_{\max}$, and also clip eq. (23) with hyperparameter of $A_{\max}$. We use *geometric noise scheduling* from Song et al. (2021); Karras et al. (2022) to schedule the noise levels $\{\sigma_t\}_{t \in [0,1]}$, controlled by hyperparameters $\sigma_{\max}$ and $\sigma_{\min}$ (See eq. (64)). The hyperparameters for each experiment are organized in Table 12.

**Comparisons and Evaluation metric** For both the MMD and Sinkhorn distance in Table 2, we follow the implementation of (Blessing et al., 2024). The benchmark results are taken from Blessing et al. (2024), Chen et al. (2024) and Akhound-Sadegh et al. (2024). For metrics in Table 3, we evaluate KL divergence ($D_{\mathrm{KL}}$) and Wasserstein distance between energy values ($E(\cdot)W_2$) of generated and reference samples. For $W_2$ estimation, we use Python OT (POT) library (Flamary et al., 2021). For all metrics, we use 2000 generated and reference samples.

