# OpenReview forum: "Non-equilibrium Annealed Adjoint Sampler"
_NeurIPS.cc/2025/Conference — NeurIPS 2025 poster_

### Official Review · Reviewer_TXNd · 2025-06-30

**Clarity:** 2
**Significance:** 3
**Originality:** 4
**Rating:** 4
**Confidence:** 4

**Summary:**

This paper proposed a novel neural sampler based on annealed path and stochastic optimal control.
Unlike previous works, which learn a single diffusion model or an interpolant-based process (e.g., based on escorted Jarzynski), this work is based on a combination of both.
More precisely, it learns two processes based on SOC. The first can be effectively trained with RAM and maps from a tractable prior to an intractable distribution. The second takes an interpolant process as the reference and maps the intractable distribution to the final target.

**Questions:**

1. Why is the ALDP exp only compared with DDS? What prior did you use for the ALDP experiment? Is the prior tuned to ensure coverage?


2. Is it easy to learn to sample from the intractable prior? As shown in Fig 1, it does not make a big difference between the prior and the target.

3. I still have difficulty intuitively understanding what makes NAAS better than other choices.

For example, intuitively, I suppose the 2nd transport (from an intractable prior to the final target) still has the teleportation issue, even though the prior is learnt, as the reference is defined by a geometric interpolant. But the vector field might be much easier compared to those defined by NETS/CMCD due to the similarity between the  intractable prior and the final target.
On the other hand, the first transport (from Gaussian to intractable prior) still has an uninformative reference. This might have an easier learning process compared to standard adjoint sampling. So can I conclude that NAAS performs better as it is an "interpolant" between AS and CMCD/NETS so that it can leverage both advantages?

However, if this is the case, I am curious why this will not be a dilemma? More precisely, to make the first transport easier, it is better to have an intractable prior similar to the Gaussian. But this will lead to a more tricky learning procedure for the second transport. Is this a concern?


4. Following the last question, if we concatenate two AS, or use $dX_t = -\nabla U_t(X_t) dt + \sqrt{2}dW_t, X_0 \sim \text{Gaussian}$ as the reference for the first process, will the results be worse than NAAS?


5. I may not fully agree with the classification in the introduction. For example, both CMCD and PIS/DDS (or other SOC based sampler) are based on KL divergence between the current controlled process and the target process running from target marginal (can be viewed as optimal control). Why are PIS/DDS "no reliance on IWS" while CMCD is not?



6. AS achieves high efficiency by decoupling gradient evaluations from direct calls to the target energy function. In contrast, NAAS requires explicit computation of the target in Eq 22. This bottleneck is also noted in recent literature, where many neural samplers are shown to underperform classical samplers like parallel tempering in terms of efficiency.  Is this also a limitation for NAAS?



**This method is interesting and I generally enjoy reading this paper. I am very happy to raise my score to accept if the above question & concerns are addressed/discussed.**

**Ethical Concerns:**

["NO or VERY MINOR ethics concerns only"]

**Final Justification:**

The rebuttal addresses most of my concerns.  I am happy with the acceptance.  I believe the method itself is interesting and is well-grounded, but as there might be many modifications to the final paper, I keep my score as 4 due to a minor concern about how well these discussions (also discussions between authors and other reviewers) can be rearranged into the final version. Especially, I believe a stronger motivation would help to clarify “this is exactly what we need to do” rather than merely “stitching together what we can do”.

**Limitations:**

The authors have adequately addressed the limitations and potential negative societal impact of their work

**Quality:**

3

**Strengths And Weaknesses:**

## Strength

1. The method is novel and inspiring. Unlike previous works that focus on one family of reference processes, this approach combines two of them, which has the potential to address the limitation of both (for example, the Jarzynski-style reference), as it has information to guidance the sampler while the other one is easy to handle as it can be trained without simulation and has no teleportation issue.

2. NAAS has promising performance compared to other baselines. It can also work on ALDP, a target that many of the current neural samplers failed without getting access to any data.

3. The writing is very clear and easy to follow.

## Weakness

1. The motivation and classification seem to be inaccurate to me. For example, the authors motivate NAAS as a sampler that does not rely on importance sampling. However, it is hard to see why IS is a clear obstacle. It is easy to see the IS in iDEM suffers; however, I am not strongly convinced that the SMC in SCLD and PDDS has similar issue. Also, I do not understand why CMCD also relies on IS.

2. I am not fully convinced why the proposed combination of two processes has advantages over other choices. Please see my questions for more details.

---

> ### Author Rebuttal · Authors · 2025-07-31
>
> We appreciate the reviewer for spending time reading our manuscript carefully and providing valuable feedback. We hope that our responses are helpful in addressing the reviewer’s concerns. Due to the strict character limit, we have grouped and summarized related comments where appropriate.
>
> ---
> **1. Clarification on motivation (W1)**
> We agree that the current presentation of our motivation should be improved. Rather than positioning our method solely as an annealing-based sampler that avoids importance sampling, we will present it as one of the key characteristics of our approach. Instead, we would like to clarify that the core motivation and contribution of our work lies in developing a SOC-based sampler that incorporates annealed reference dynamics, as illustrated in the following flow:
>
> - Prior SOC-based approaches [1,2,3] have constructed unbiased diffusion samplers using stationary or uninformative base references, and do not fully leverage annealing references as employed in other importance sampling based methods. In this work, we propose a novel SOC-based framework that employs annealed reference dynamics as the base SDE.
> - Precisely, we formulate two SOC subproblems: (i) learning a prior distribution $\mu$ from the initial distribution $\delta_0$ with standard base reference, and (ii) transporting $\mu$ to the target $\nu$ using controlled annealed dynamics. We unify these into a single SOC formulation, which yields a controlled diffusion process that effectively transports $\delta_0\to\mu\to\nu$.
> - Among available solvers for SOC, we adopt Adjoint Matching [3], a recently proposed method being scalable and sample-efficient. As discussed in Adjoint Sampling (AS) [4], this approach enables scalability and improved training stability.
>
> ---
> **2. Clarification on importance sampling (W1)**
>
> As established in prior works (e.g. [5]; [6] Section 4.1.4), importance sampling can suffer from high variance in high-dimensional settings, particularly when estimating expectations under complex distributions. In such cases, one must compute unnormalized weights $\\{w^{(i)}\\}^N_{i=1}$ corresponding to sampled trajectories $\\{X^{(i)}\\}^N_{i=1}$, where $$w^{(i)}\propto\frac{dp^\star}{dp}(X^{(i)}),$$ where $p^\star$ denotes the target path measure, while $p$ is the proposal. These weights are then used to form an empirical approximation of the target distribution,$$\nu\approx\sum^N_{i=1}w^{(i)}\delta_{X^{(i)}}.$$
>
> When the support or mass of the proposal and target distributions are misaligned, as is common in high-dimensional or complex energy landscapes. This leads to poor effective sample size and significant variance in the empirical approximation, which is further exacerbated when the number of samples $N$ is insufficient. For example, in PDDS [7], the learned distribution approximates a reweighted empirical measure and is thus exposed to the instability induced by high-variance weights. In contrast, our method learns a sampling distribution directly via path-space control, thereby avoiding the need to compute or rely on importance weights.
>
> ---
> **3. Clarification on combination of two stochastic processes (W2, Q2, Q3, Q4)**
>
> **3.1 The role of two optimization stages (W2, Q2)**
>
> We appreciate the reviewer for raising this important point. We agree that we need a more intuitive explanation of the role of two optimization problems. We incorporate the following intuitive explanation in our main manuscript in the last part of Section 3.
>
> The two optimization problems in our framework, (i) learning the prior distribution $\mu$ over the interval [−1,0], and (ii) optimizing the controlled annealed dynamics over [0,1], exhibit fundamentally different learning dynamics and objectives.
>
> The first-stage optimization over [−1,0] learns a prior distribution $\mu=e^{-U_0-V_0}$, where $U_0$ is a simple, prescribed energy function (typically corresponding to Gaussian distribution). This prior is designed to have broad and smooth support, in contrast to the often sharp or multimodal structure of the target distribution $\nu$. The objective here is not to capture fine-scale structure, but to construct a low-complexity distribution that can be easily sampled using a simple base process, while still broadly covering the support of $\nu$.
>
> In contrast, the second-stage optimization over [0,1] focuses on refining the annealed dynamics to guide trajectories from $\mu$ toward the target distribution $\nu$. Since the annealed reference dynamics already bias the flow toward high-density regions of the target, the control at this stage primarily serves to correct residual mismatches. Because we only required to learn the residual mismatch, the second optimization problem is also easy to learn.
>
> By decoupling these two objectives, the alternating scheme enables more stable and effective training. Each optimization stage solves a simpler and more focused subproblem: the first constructs a well-behaved initialization, and the second incrementally corrects the flow toward the target. This separation reduces optimization interference and improves both convergence and sample quality.
>
> **3.2 Explanation of two subproblems and the extreme case (Q3)**
>
> Thank you for pointing out the important issue that “NAAS performs better as it is an interpolant between AS and Annealed-based Methods”. We largely agree with this intuition, and we elaborated it on the previous paragraphs. In short, the first optimization term is AS [3] with easier target $\mu$, and the second optimization term is correction of the annealed dynamics.
>
> This decomposition allows each stage to focus on a well-scoped subproblem and mitigates the difficulty of learning a single controlled dynamics. As the reviewer noted, this design introduces a natural tradeoff: If $\mu$ is very close to a standard Gaussian, the learning burden is shifted to the second stage, resembling an annealed SOC approach. Conversely, if $\mu$ is close to $\nu$, the second stage becomes nearly trivial, reducing the method to something closer to AS [3]. But, anyhow, compared to both AS [3] and annealing-based methods, NAAS consistently gains from partitioning the task into structured subproblems.
>
> We thank the reviewer for this valuable insight. We will include a discussion of investigating this trade-off as a promising avenue for future work in the conclusion of the revised manuscript.
>
> **3.3 Why not use AS twice or annealed dynamics for 1st segment (Q4)**
>
> Conceptually, one might consider alternative designs, such as stacking two AS stages or employing annealed dynamics in the prior learning stage. However, our method is deliberately designed to ensure unbiased sampling, which imposes structural constraints on how the reference dynamics are chosen in each stage.
>
> In the first optimization stage (over time interval [-1,0]), the goal is to generate a sample proportional to the prior distribution $\mu$ via our controlled dynamics. To guarantee unbiasedness of our controlled dynamics, the reference dynamics in this stage must be memoryless, like PIS [1] and DDS [2] as described in L97-115. Annealed dynamics violate this memoryless condition, and therefore cannot serve as a valid reference in this stage. Consequently, we adopt standard Brownian motion as the reference dynamics during [-1,0].
>
> Additionally, to empirically validate the effectiveness of using annealed dynamics as a reference in the second stage (over [0,1]), we compare our method against AS [3], which shares the same SOC formulation but uses a static base. As shown in the table below, our method achieves significantly better performance highlighting the practical benefit of the annealed reference dynamics. We attribute this improvement to the fact that the annealed dynamics guide samples toward high-density regions of the target distribution, thereby facilitating the learning of optimal control.
>
> |Sinkhorn| MW54 | GMM40 | MoS|
> |----|----|----|----|
> |AS|0.32$\pm$0.06|18984.21$\pm$62.12|2178.60$\pm$54.82|
> |Ours|**0.10**$\pm$0.0075 |**496.48**$\pm$27.08|**394.55**$\pm$29.35|
>
> ---
> **4. Clarification on remaining questions (Q1,Q5,Q6)**
>
> **Q1. ALDP&DDS.** Thank you for the question. In addition to DDS [2], we also implemented and evaluated several SOC-based samplers, including PIS [1] and LV-PIS [11], on the ALDP experiment. However, we couldn’t obtain converging result for LV-PIS. We provide the full comparison results in the table below:
>
> |Method|$\phi$|$\psi$|$\gamma_1$|$\gamma_2$|$\gamma_3$|E$W_2$|
> |----|----|----|----|----|----|----|
> |PIS|0.597 |0.952|0.453|0.498|7.038|5.918|
> |DDS|0.493 |0.154|4.095|0.111|0.150|5.467|
> |Ours|0.260|0.236|0.272|0.132|0.156|**1.076**|
>
> As shown in the table, our method (NAAS) achieves the best performance across most of the evaluated metrics. Regarding the prior, we learn the prior distribution $\mu$ through a dedicated SOC problem over the interval [-1,0].
>
> **Q5. CMCD and IS.** We apologize for misleading you and thank you for helping us identify this important mistake. We should’ve put CRAFT [8] as a reference instead of CMCD [10]. We modified the manuscript.
>
> **Q6 AS.** We agree that NAAS incurs additional computational cost compared to methods like AS [3], due to the need to compute the adjoint state via an ODE and evaluate gradients of the energy function during training (see Eq. (22)). This computational overhead is indeed a limitation. We added a discussion of this limitation to the conclusion of our manuscript.
>
> ---
> References
>
> [1] Path Integral Sampler
>
> [2] Denoising Diffusion Samplers
>
> [3] Adjoint Sampling
>
> [4] Adjoint Matching
>
> [5] Sequential Monte Carlo Samplers
>
> [6] Free energy computations: A mathematical perspective
>
> [7] Particle Denoising Diffusion Sampler
>
> [8] Continual Repeated Annealed Flow Transport Monte Carlo
>
> [9] Sequential Controlled Langevin Diffusions
>
> [10] Controlled Monte Carlo Diffusions
>
> [11] Improved Sampling via Learned Diffusions

---

> > ### Comment · Reviewer_TXNd · 2025-08-02
> >
> > Thank you for your reply and classification. However, i may still have a few questions & disagreement:
> >
> > > Among available solvers for SOC, we adopt Adjoint Matching [3], a recently proposed method being scalable and sample-efficient. As discussed in Adjoint Sampling (AS) [4], this approach enables scalability and improved training stability.
> >
> > I agree AS is efficient and scalable due to its RAM objective, which does not require simulation for each grad update. However, in this work, only the first process makes use of this property. Therefore, I am not fully convinced that this approach has the same advantage.
> >
> >
> > > to empirically validate the effectiveness of using annealed dynamics as a reference in the second stage (over [0,1]), we compare our method against AS [3], which shares the same SOC formulation but uses a static base
> >
> > Thank you for these additional experiments. However, did this AS use a single process from the prior to the target? If so, it does not quite answer my question. What i am curious is: is the gain due to the additional process/network which enables higher flexibility, or its due to this specific design?

---

> > > ### Author Response · Authors · 2025-08-04
> > >
> > > We appreciate the reviewer’s prompt response and the important questions. Below, we elaborate on our replies and provide some clarifications:
> > >
> > > ---
> > > > I agree AS is efficient and scalable due to its RAM objective, which does not require simulation for each grad update. However, in this work, only the first process makes use of this property. Therefore, I am not fully convinced that this approach has the same advantage.
> > >
> > > We thank the reviewer for raising the comments. As the reviewer conjectured, our NAAS indeed only makes use of the computational advantages of AS in its first process, between [-1,0], while employing full AM in the second process, between [0,1], with a much more informative annealing reference. This significantly improves the performance, as evidenced by Table 2 and the table below, despite making it less efficient compared to AS. We acknowledge that current revision may cause confusion, and will revise the writing accordingly.
> > >
> > > We note, however, that employing full AM in the second process still possesses efficiency over other alternatives, such as naive back-propagation through the SDE simulation graph and LogVar [11]. Specifically, for the LogVar, one would have to implement their “bridge” formulation (see their Table 1) due to the nontrivial prior. This necessitates training additional backward control from t=1 to 0, which can be computational expansive, *and* knowing $B(X) = \log \frac{\mu(X_0)}{\nu(X_1)}$, which is intractable for our prior $\mu = e^{-U_0 - V_0}$ due to the absence of intractable $V_0$.
> > >
> > >
> > > ---
> > > > Did AS use a single process from the prior to the target? If so, it does not quite answer my question. What i am curious is: is the gain due to the additional process/network which enables higher flexibility, or its due to this specific design?
> > >
> > >
> > > We appreciate the reviewer for raising the question. Following the reviewer’s suggestion, we conduct additional experiments to verify whether the improved performance of NAAS over AS arises from the use of annealing reference or, as conjectured by the reviewer, from the flexibility from the second process/network. Specifically, we instantiate NAAS with a particular choice of $U_t$, which remains equal to $U_0$ for all $t \in [0,1)$ and jump to $U_t = U_1$ at terminal $t=1$. This setup effectively leads to a two-staged AS baseline that retains the same first process [-1,0] as NAAS, while completely removing the annealing reference in the second stage. As we proved in Corollary 3.2., the optimal process of this two-staged AS baseline satisfies the desired target. In other words, any performance gaps between this ablated baseline and NAAS can be attributed to the use of annealing reference paths.
> > >
> > >
> > > We report the preliminary results in the table below, where the two-stage AS (first row) performs similarly to the original AS. In contrast, our NAAS (second row) outperforms this baseline by a significant margin. These results highlight the important role of the annealed reference in NAAS, which contributes non-trivially to its improved performance. We will include this discussion in the revision and thank the reviewer again for raising this insightful question.
> > >
> > > |Sinkhorn($\downarrow$)| MW54| GMM40|
> > > |----|----|----|
> > > | $U_{t \in [0,1)} = U_0$ and $U_{t=1} = U_1$ | 0.43 | 19776.91 |
> > > | $U_t = (1-t)U_0 + t U_1$ (Ours) | 0.10 | 496.48 |
> > >
> > > ---
> > >
> > > Finally, we note that the advantage of NAAS over AS should not be merely understood through an engineering perspective, because the underlying idea is fundamentally advantageous. In particular, we used a time horizon of [-1,1]. Its [0,1] component is better than AS because it uses a much better reference process, and its [-1,0] component is better than AS because it has a much easier target.

---

> > > > ### Comment · Reviewer_TXNd · 2025-08-04
> > > >
> > > > Thank you for your reply! i have increased my score

---

### Official Review · Reviewer_GpP4 · 2025-07-02

**Clarity:** 3
**Significance:** 2
**Originality:** 2
**Rating:** 4
**Confidence:** 5

**Summary:**

This paper proposes a novel variational diffusion-based sampler defined on time interval $[0,1]$, that aims to sample at $t=1$ from an unnormalized target density $\pi$ (without access to ground truth samples), by leveraging "reference" stochastic dynamics informed by the target distribution. The goal is to learn an additive control to the "reference" drift, to steer the particles towards the high density areas (which are already given by the reference dynamics) but with the right proportions (which is the harder task). In line with recent works [1, 2], the authors consider the Annealed Langevin dynamics as reference, where the time-evolving density $p_t^{\text{ref}}$ is a geometric interpolation between a standard Gaussian distribution and $\pi$. However, the formulation of the problem is different : while previous works meant at correcting the reference dynamics to marginally match $p_t^{\text{ref}}$ via a variational formulation, the current approach relies on a novel Stochastic Optimal Control (SOC) formulation which enables to exactly match $\pi$ at time $t=1$ but not $p_t^{\text{ref}}$ for $t<1$. In particular, the induced SOC loss relies on a state cost and not a final cost, as opposed to previous (memoryless) SOC-based diffusion samplers, and can be optimized via the backward Adjoint Matching procedure [3]. In this specific case however, the starting marginal at time $t=0$ is intractable and needs to be approximated to obtain starting samples. To circumvent this issue, the authors propose to extend the SOC to $[-1,0]$, by formulating a standard memoryless setting (ie denoising diffusion), which can be solved via the Adjoint Sampling method [4]. In the end, the overall algorithmic procedure alternates between minimizing these two objectives (hence learning two different neural networks), which can be made by sharing the use of buffers. This method is compared with annealed MCMC methods and recent diffusion-based samplers on toy mixture settings and molecular systems.

[1] Transport meets Variational Inference: Controlled Monte Carlo Diffusions. Vargas et al, 2024

[2] Sequential Controlled Langevin Diffusions. Chen et al. 2025

[3] Adjoint Matching: Fine-tuning Flow and Diffusion Generative Models with Memoryless Stochastic Optimal Control. Domingo-Enrich et al. 2025

[4] Adjoint Sampling: Highly Scalable Diffusion Samplers via Adjoint Matching. Havens et al 2025.

**Questions:**

**Questions**
- Following my remark in the Section *Weaknesses* on Figure A, where the trickiest part of the multimodal problem seems to be solved in the first interval $[-1,0]$, how can you handle in practice the tradeoff of "hardness" (i.e. controlling carefully the mode speciation) between the two SOC parts ? Does it relate to the setting of $\sigma_t$ ? Can it rely on realistic hyperparameter tuning (ie without access to ground truth samples) ?
- How do you initialize the buffers in practice ? This is often the tricky part in diffusion samplers. For instance, LRDS leverages the existence of Monte Carlo samples collected in the target modes (but not informed by the relative weights) to initialize the reference drift; AS also uses this technique to evaluate for the first time its variational loss. These approaches hence ensure fast convergence. In the proposed setting, I do not see how such target-informed samples could be used due to the fact that we have no information on the marginal distribution at time $t=0$, which may lead to slow convergence. Do I miss something ?

**Suggestions**
- one of the main motivations of this work is to avoid the use of importance sampling schemes, which seems to be denoted as something negative. Although I understand that indeed, IS can lead to high variance when the target and the proposal do not match, even more when the dimension is large; however, resampling is not necessarily doomed to fail in general settings, if the target and the proposal are well designed. Most of real-world samplers rely on versions of Parallel Tempering, where the resampling steps work because the intermediate temperatures are carefully chosen to avoid huge mismatch between densities. Moreover, resampling may help to recalibrate the diffusion model with respect to the target distribution, hence avoiding accumulating too much numerical bias when generating from the neural sampler (especially at the very end, where we want to be sure to sample from $\pi$). I think that this motivation should be more nuanced.
- I recommend the authors to consider multi-modal targets with non-uniform mode weights where the measurement of mode weight recovery (which is non-trivial at all for diffusion-based samplers) can be explicitly computed, see for example [1]. This would enable to understand if the proposed sampler suffers from mode switching and/or mode blindness issues (as mode discovery is already partially handled by the reference dynamics).
- the benchmark should include (i) LRDS, which is the natural competitor of the current approach for a memoryless setting and (ii) AS, to understand the improvements of using the reference Annealed Langevin dynamics in the second SOC part (more informally, can AS already do the job ?)
- following my last point, I think it would be interesting to have a visual inspection of the marginal distribution at time $t=0$ depending on several hyperparameter settings, to understand if it already captures the multimodality features of the target, and therefore if the use of annealed reference dynamics is that useful

[1] Improving the evaluation of samplers on multi-modal targets. Grenioux et al, 2025.

**Ethical Concerns:**

["NO or VERY MINOR ethics concerns only"]

**Final Justification:**

I have increased my score, as the authors have provided convincing answers that will be added to the paper (see my final answer for more details).

**Limitations:**

To apply the Adjoint Matching method, one needs to have access to the Hessian of $\log\pi$, which may be costly to derive; plus, one needs to train two neural networks in the end. This computational burden should be much more emphasized as a limitation of the current method in a context where the computational budget of diffusion-based samplers is questioned, see [1].

[1] No Trick, No Treat: Pursuits and Challenges Towards Simulation-free Training of Neural Samplers. He et al, 2025.

**Quality:**

3

**Strengths And Weaknesses:**

**Strengths**
- the paper is technically solid, the introduction of the losses is well detailed and carefully justified
- the paper is overall well written, the figures explain well the main ideas

**Weaknesses**
- the current paper does not cite a close competitor LRDS [1] (which was publicly available more than 6 months ago), which aims at solving the same kind of task (i.e., learning a diffusion based sampler by leveraging reference dynamics that is informative on the target without introducing importance sampling scheme) in the memoryless setting (ie with denoising diffusion), while focusing on multi-modal targets. This competitor should be compared with the current method to weigh the pros/cons between the two approaches (in particular, LRDS only learns one control term): namely, should we consider a reference dynamics given by a denoising or a tempering path ?
- the current method should also be compared with the Adjoint Sampling method in the numerics (ie the classic memoryless setting, yet with non-informed reference dynamics), to understand how incremental the improvement is by allowing the use of reference dynamics with tempering path (which increases the problem complexity). In particular, the results of Figure 1 might suggest that **the mode speciation** of the target density (ie the trickiest part when sampling from multimodal targets starting from unimodal distributions) **may already happen on $[-1,0]$** (ie when applying AS method), resulting in separated modes in the AM part : in this specific case, the annealed reference dynamics may not be much necessary (applying Langevin may be sufficient to capture the local landscape)...
- the issue of *mode switching*, which is inherent to annealed sampling, is not discussed : it relates to the fact that the mode weights may vary a lot on a given tempering path (which is not the case for denoising paths), which is one of the motivations stated in [1] ti use a denoising path. This may notably result in obtaining wrong mode proportions (while still good mode locations) by the end of the sampling. This has notably motivated previous work [2] to learn the intermediate temperature levels, to be able to suffer less from mode switching (see their Figure 8). Here, the choice of the toy multimodal targets presented in the numerics does not enable to see if the presented method avoids this phenomenon (which may definitely happen since the chosen geometric interpolation is the linear one and is not optimized). Remark that the sampling results on GMM40 (see Figure 2.a) seem to indicate that the true mode proportions are indeed not well recovered by the algorithm...
- in the same manner, the issue of the *mode blindness of score-based losses* (here, the AS loss on $[-1, 0]$) is not addressed: in the case of well separated modes, the score of the target density is known to be blind to the mode weights (contrary to the log density), see [3]. Hence, for a multimodal target with non equilibrated modes, this blindness of $\nabla \log \pi$ may prevent to reach the target when using AS methodology and rather end up with obtaining samples from an equilibrated version of $\pi$ (same mode locations, but wrong mode proportions). The current target settings do not enable to observe if the proposed sampler solves this issue.
- while the proposed approach seems to rely on careful hyperparameter tuning (namely $\sigma_t$), which, as far as I understand, may be critical to split the task hardness between the two SOC problems (see the section *Questions*). No indication of setting in realistic sampling frameworks is provided in the main paper, which hurts the applicability to real word problems

[1] Learned Reference based Diffusion Sampler for multi-modal distributions. Noble et al, 2025.

[2] Sequential Controlled Langevin Diffusions. Chen et al. 2025

[3] Towards Healing the Blindness of Score Matching. Zhang et al. 2022.

---

> ### Author Rebuttal · Authors · 2025-07-31
>
> We appreciate the reviewer for spending time reading our manuscript carefully and providing valuable feedback. We hope that our responses are helpful in addressing the reviewer’s concerns. Due to the strict character limit, we have grouped and summarized related comments where appropriate.
>
> ---
> **S1**. One of the main motivations of this work is to avoid the use of importance sampling schemes.  Although I understand that indeed, IS can lead to high variance when the target and the proposal do not match, even more when the dimension is large; however, resampling is not necessarily doomed to fail in general settings, if the target and the proposal are well designed. Moreover, resampling may help to recalibrate the diffusion model with respect to the target distribution, hence avoiding accumulating too much numerical bias when generating from the neural sampler.
>
> **A**. We thank the reviewer for raising an important point regarding the role of importance sampling in our motivation. We agree that the current presentation lacked sufficient justification and clarity on this aspect. In response, we have revised the introduction to better highlight our core contribution: the development of an SOC-based sampler with annealed reference dynamics, which is a strong base dynamics that facilitates transport toward the target distribution. And among various solver of the SOC-based sampler, we employ Adjoint Matching [4], which is known to be scalable, has low-variance and efficient [3]. Precisely, we will revise the introduction to reflect the following structure:
>
> > Prior SOC-based approaches such as PIS [1], DDS [2], and AS [3] aim to solve the SOC formulation (e.g., Equation (8) or (9)) to construct unbiased diffusion samplers. However, these methods typically learn the controlled dynamics starting from a stationary or uninformative base reference, and therefore do not fully leverage annealing references as employed in other importance sampling based methods.
>
> > Motivated by this limitation, we propose a new SOC-based framework that employs annealed reference dynamics as the base SDE. This annealing structure provides a natural progression toward the target distribution and generates more informative reference trajectories, thereby significantly improving the stability and efficiency of learning the control. As we demonstrate empirically, this design leads to improved sample quality and target matching, especially compared to methods that rely on static references.
>
> > To delve deeper into our method, we formulate two SOC subproblems: (i) learning a prior distribution $\mu$ from the initial distribution $\delta_0$ with standard base reference, and (ii) transporting $\mu$ to the target $\nu$ using controlled annealed dynamics. We unify these into a single SOC formulation, which yields a controlled diffusion process that effectively transports $\delta_0 \to \mu \to \nu$, resulting in a sampler whose terminal distribution matches the target.
>
> > Among available solvers for SOC, we adopt Adjoint Matching [4], a recently proposed method being scalable and sample-efficient. As discussed in Adjoint Sampling [3], this approach enables low-variance and improved training stability.
>
> > To summarize, we introduce a novel SOC-based diffusion sampling framework that leverages annealed reference dynamics to enhance learning of the control. This framework, which we call NAAS (Non-equilibrium Annealed Adjoint Sampler), is the first to formulate an SOC problem over diffusion processes with a non-stationary, annealed base. Our approach combines principled formulation with a practical and scalable solver, and demonstrates strong empirical performance across several challenging sampling tasks.
>
>
> ---
> **W1-5, S2**. Consider multi-modal targets with non-uniform mode weights where the measurement of mode weight recovery. This would enable to understand if the proposed sampler suffers from mode switching and/or mode blindness issues. The benchmark should include (i) LRDS (ii) AS, to understand the improvements of using the reference Annealed Langevin dynamics in the second SOC part.
>
> **A**. We thank the reviewer for the thoughtful and constructive suggestions. We fully agree on the importance of evaluating whether the proposed sampler can accurately recover mode weights in multi-modal distributions with non-uniform weights. As suggested, we adopt the experimental setup introduced in LRDS [2], focusing on a two-mode Gaussian mixture model (GMM) where the first mode has a true weight of $w_1=2/3$​. We report the absolute error $|\hat{w}_1-w_1|$, where $\hat{w}_1$​ is the empirical weight estimation of the first mode.
>
> | Method ($\downarrow$) | $d=16$ | $d=32$ |
> |----|----|----|
> |AS| 20.5\% | - |
> |LRDS| **1.7**\% | 2.7\% |
> |Ours| 3.3\% | **2.0**\% |
>
> As shown in the table, our method demonstrates competitive performance and recovers the correct mode weights with high accuracy. This indicates that our method does not suffer from mode blindness or switching in this scenario. Interestingly, we observed that the controlled dynamics in the first stage (from time $[-1,0]$ plays a key role in reallocating the weight. At the beginning of the training, $\hat{w}_1 \approx 0.5$, but as we learn the control for time $[-1,0]$, it gradually reallocates weights towards $\hat{w}_1 \approx 2/3$.
>
> We also note that both AS and our method converged stably with a small learning rate of $10^{-6}$. However, due to the limited rebuttal period, we could not perform an extensive hyperparameter search for AS, which may partially explain its degraded performance. We plan to conduct a more comprehensive hyperparameter search for AS and include the results in the revised version of the manuscript.
>
> To further verify the benefit of incorporating annealed dynamics, we additionally applied AS to the main benchmark tasks from Table 2. As shown below, our method consistently outperforms AS by a significant margin, further supporting the effectiveness of annealed reference dynamics:
>
> | Sinkhorn | MW54 | GMM40 | MoS|
> |----|----|----|----|
> | AS | 0.32 $\pm$ 0.06 | 18984.21 $\pm$ 62.12 | 2178.60 $\pm$ 54.82 |
> | Ours | **0.10** $\pm$ 0.0075 | **496.48** $\pm$ 27.08 | **394.55** $\pm$ 29.35 |
>
>
> ---
> **W6, S3, Q1**. Impact of hyperparameter, especially $\sigma_t$ (W6). It would be interesting to have a visual inspection of the marginal distribution at time $t=0$ depending on several hyperparameter settings (S3). Following my remark in the Section Weaknesses on Figure A, where the trickiest part of the multimodal problem seems to be solved in the first interval [-1,0], does it relate to the setting of $\sigma_t$ (Q1)?
>
> **A**. Regarding the visual inspection of the marginal distribution at $t=0$, due to rebuttal policies, we are currently unable to include the visualizations. However, we have verified that increasing the guidance strength (i.e., larger $\sigma_t$​) causes the distribution $\mu$ at $t=0$ to resemble the reference distribution$\propto e^{-U_0}$​, which aligns with equation (13). We will include these visualizations and a more detailed discussion in the revised version of the manuscript.
>
> ---
> **W7**. No indication of setting in realistic sampling frameworks is provided in the main paper, which hurts the applicability to real word problems
>
> **A**. We agree that evaluating our method on more realistic, large-scale applications (such as molecular generation) would further enhance its practical applicability. However, we would like to kindly note that our current experiments readily include challenging, high-dimensional problems in sampling literature [3,4,7,8] where ground-truth samples are obtainable. Specifically, we include the GMM40 and MoS datasets in Table 2 which are 50 dim, and the alanine dipeptide in Table 3 has 60 dim.
>
> ---
> **Q2.** How do you initialize the buffers in practice? I do not see how such target-informed samples could be used due to the fact that we have no information on the marginal distribution at time $t=0$, which may lead to slow convergence.
>
> **A**. We do not use any special initialization for the buffers. For each epoch, we generate a certain number of trajectories (in most of our experiments, we used 512 trajectories) sampled from the range [-1, 1]. As the reviewer correctly noted, we do not have access to the prior distribution $\mu$ at time $t = 0$. To address this, we obtain samples by starting from $X_{-1}\sim\delta_0$. Please note that adjoint-based methods only requires (on-policy) samples, so there is no need to get correct samples at $t=0$.
>
> ---
> **L1** Limitations Several drawbacks: (1) requires access to the Hessian of $U_t$ to apply Adjoint Matching, (2) requires two neural networks (see [6]).
>
> **A**. We agree with the reviewer that we should discuss the limitation that pointed out. We will incorporate it in the conclusion section.
>
> ---
> Once again, we appreciate the reviewer for careful consideration and important suggestions. We hope that our responses are helpful in addressing the reviewer’s concerns.
>
>
> ---
> **References**
>
> [1] Improving the evaluation of samplers on multi-modal targets. Grenioux et al, 2025.
>
> [2] Noble et al., Learned Reference-based Diffusion Sampling for Multi-Modal Distributions, ICLR, 2025.
>
> [3] Havens et al., Adjoint Sampling: Highly Scalable Diffusion Samplers via Adjoint Matching, ICML, 2025.
>
> [4] Chen et al., Sequential Controlled Langevin Diffusions, ICLR, 2025.
>
> [5] Towards Healing the Blindness of Score Matching, Zhang et al., 2022.
>
> [6] No Trick, No Treat: Pursuits and Challenges Towards Simulation-free Training of Neural Samplers. He
> et al, 2025.
>
> [7] Improved Sampling via Learned Diffusions
>
> [8] Non-equilibrium Transport Sampler

---

> > ### Comment · Reviewer_GpP4 · 2025-08-05
> > **Answer to the rebuttal**
> >
> > Thank you for taking into account my suggestions and providing a detailed response to my review. I will answer to your rebuttal point by point.
> >
> > **Answer to S1.** Than you a lot for clarifying the motivation of your approach, which clearly focuses on providing meaningful reference dynamics for the diffusion-based sampler as in LRDS and therefore, avoids use of intensive resampling.
> >
> > **Answer to W1-5, S2..** Than you for the additional experiment and integrating AS in your benchmark (which clearly show the increment brought by your method). To bring even more consistency, I would suggest to the authors to study settings where the dimension and/or the between-mode distance increases, to inspect the robustness of the current approach in challenging multi-modal settings (which is the core objective of the paper), see [1] for instance. I still have questions related to this:
> > - do you ensure a fair computational budget between LRDS and your method ? (one neural network vs 2 neural networks)
> > - did you also integrate LRDS in your benchmark to ensure fair comparison (since they both rely on the same principle) ?
> > - the indicated learning rate $10^{-6}$ for your approach seems really low; do you observe loss instability when training compared to LRDS for instance, or difference in convergence speed ?
> > - about the mode switching behaviour : for arbitrary multi-modal targets with unbalanced modes, the linear geometric path will likely suffer from this phenomenon, which may bring bring difficulty in the learning process. I am afraid that the limited benchmark proposed in the paper does not reflect it. Do you have a strategy to mitigate this issue ?
> >
> > **Answer to W6, S3, Q1.** Thank you for clarifying the role of $\sigma_t$. This is for me the most critical hyperparameter of the method, which definitely controls how the difficulty of the sampling problem is spread between the two time intervals. Do you observe difference in performance when using different values ? If yes, an ablation study on its role wrt the sampler performance should definitely be given. Honestly, I do not see any practical way to tune it without having access to ground truth samples as validation, and therefore it acts as a weakness in my opinion. Do you have strategy to set it in practice ?
> >
> > **Answer to Q2.** So if I understand well, you initialize the procedure by sampling trajectories from time $t=-1$ using a zero-valued neural networks ? What happens if the hyperparameter $\sigma_t$ is not correctly set, could you miss modes of the target and suffer from a intrinsic bias ? In this uninformative case, how does the convergence speed compare with LRDS (which can already sample meaningful points at the very beginning) ?
> >
> > [1] Improving the evaluation of samplers on multi-modal targets. Grenioux et al, 2025.

---

> > > ### Author Response · Authors · 2025-08-08
> > > **Author response to reviewer's comments (part 1/2)**
> > >
> > > We thank the reviewer for the reply and are excited that the reviewer appreciated our clarification. We provide additional clarification and experiments below.
> > >
> > > ---
> > >
> > > **1. Additional experiments on multi-modal**
> > >
> > > - To further examine the robustness of NAAS in challenging multi-modal settings, we conduct ablation studies on the same mode weight estimation problem as reported in our initial rebuttal, but with an increasing between-mode distance. Specifically, we consider the bi-modal Gaussian mixture with components located at $\mathcal{N}(a\mathbf{1}_d, \Sigma_1)$, with weight $w_1 = 2/3$, and $\mathcal{N}(a\mathbf{1}_d, \Sigma_2)$, with weight $w_2 = 1/3$. Notice that setting $a=1$ recovers the setup we previously reported. To ensure a fair comparison, we set the network size of LRDS to be twice that of our NAAS model, so that the total number of learnable parameters is approximately equal (since NAAS employs two networks).
> > >
> > > - The table below reports our preliminary results. Notably, our NAAS achieves comparable performance to LRDS, consistently recovering the correct mode weights as the between-mode distance $a$ increases. For LRDS, we kept their default hyperparameters unchanged but trained with longer epochs on $a=2,3$ to ensure convergence. That said, we believe that performance of LRDS might be further improved with hyperparameter tuning. We emphasize that these results were _not_ intended to compare extensively between NAAS and LRDS, but rather to ablate whether NAAS remains robust in challenging multi-modal problems with widely separated modes, for which we provide positive evidence.
> > >
> > >   | $\|\hat{w}_1 - w_1\|$ ($\downarrow$) | $a=1$ | $a=2$ | $a=3$ |
> > >   |---|---|---|---|
> > >   | LRDS | 1.7% | 6.5% | 6.7% |
> > >   | NAAS | 3.3% | 3.2% | 3.6% |
> > >
> > > ---
> > >
> > > **2. Additional experiments on multi-modal**
> > >
> > > - Next, we provide results of LRDS on the benchmarks from Table 2. Consistent with the aforementioned experiments, we maintain the network size of LRDS twice that of NAAS for fair comparison. Our results indicate that LRDS stands strongly among other baselines, with the performance close to our NAAS. We will integrate LRDS with other benchmarks in the revision, and we thank the reviewer for their suggestions.
> > >
> > >   | Sinkhorn ($\downarrow$) | MW54 | Funnel |
> > >   |----|----|----|
> > >   | LRDS | 0.85 | 152.09 |
> > >   | NAAS | 0.10 | 132.30 |
> > >
> > > ---
> > >
> > > **3. Ablation studies on effect of $\sigma_t$**
> > >
> > > - As conjectured by the reviewer, $\sigma_t$ is indeed the key hyperparameter of our method, and we agree that an ablation study on the effect of $\sigma_t$ would help further demonstrate the robustness of our method. In the table below, we ablate the performance of NAAS on the GMM40 benchmark by gradually increasing the $\sigma_max$ that defines $\sigma_t := \sigma_\min^t \sigma_\max^{1-t}\sqrt{2\log \frac{\sigma_\max}{\sigma_\min}}$.
> > >
> > >   | Sinkhorn ($\downarrow$) | $\sigma_\max$=1 | 10 | 20 | 50 | 100 | 200 |
> > >   |---|---|---|---|---|---|---|
> > >   | NAAS | 22893.72 | 1350.76 | 639.86 | 496.48 | 443.55 | 492.83 |
> > >
> > > - It is evident that the performance of NAAS depends on $\sigma_\max$, as the hyperparameter directly affects its exploration capability. However, once $\sigma_\max$ surpasses some threshold (~20 for this specific benchmark), the performance of NAAS stabilizes and becomes insensitive to further increase of $\sigma_\max$. That said, tuning $\sigma_t$ in practice could be straightforwardly performed with e.g., grid or binary search, _without_ having access to samples from or near target distribution. We emphasize these results as strong evidence of the robustness of NAAS for practical usages.
> > >
> > > - While the aforementioned results have already shown robustness of NAAS w.r.t. $\sigma_t$, we would like to highlight an intriguing extension of NAAS by pretraining its control networks using samples near the target distribution. Specifically, one can compute the same bridge matching (BM) target $\alpha_t (X_1 - X_t)$ proposed in AS, with $X_1$ near-target samples, $t$ sampled uniformly from [-1,1], $X_t \sim q(X_t | X_1)$ sampled from the conditional distribution of Brownian reference paths, and $\alpha_t$ some time scaling. Then, the pretraining involves regressing $|| v^\theta_t(X_t) - \alpha_t (X_1 - X_t) ||^2$ for the first process $t \in [-1, 0]$ and $|| -\frac{\sigma_t}{2} \nabla U_t(X_t) + u^\theta_t(X_t) -  \alpha_t (X_1 - X_t) ||^2$ for the second process for $t \in [0,1]$.
> > >
> > > - While the aforementioned pretraining scheme does not necessarily lead to the solution of NAAS, it nevertheless serves as a _warm-up_ to ensure that the model can generate meaningful samples right from initialization—similar to how AS and LRDS benefit from “reference samples” near the target before proceeding with full variational optimization. We will include these discussions and the above ablation studies in the revision, and we are grateful for the reviewer’s feedback, which has helped strengthen the empirical soundness of our method.

---

> > > > ### Author Response · Authors · 2025-08-08
> > > > **Author response to reviewer's comments (part 2/2)**
> > > >
> > > > **4. Additional clarifications and discussions**
> > > >
> > > > - **Learning rate of NAAS.** We emphasize that the value we used (1e-6) is consistent with prior implementations of Adjoint Matching (AM). Specifically, the official AM codebases [9] set the default learning rate to 3e-6, which is within the same order of magnitude as that used in our experiments. That said, it is a standard practice in AM to set a smaller learning rate, and our NAAS—as an application of AM for Boltzmann sampling—follows this convention. For the completeness, we note that such a practical choice is motivated by the “self-consistency” property of the AM objective, $|| u^\theta(X_t, t) + \sigma_t a_t(X) ||^2$, where both the model’s input $X_t$ and output $-\sigma_t a_t(X)$ depend on the model itself $X \sim p^{u^\theta}$. Such consistency-based objectives generally require smaller learning rates to ensure stable training. For instance, consistency models [10] report 5e-6 (see their Appendix B.3, page 25), aligning with the values used in our work.
> > > >
> > > > - **Linear geometric path $U_t$.** This is a great question. We fully agree with the reviewer that the linear geometric path, $U_t = I(t, U_0, U_1) = (1-t) U_0 + t U_1$, used in NAAS (see L209-210) might be insufficient for more challenging sampling problems, even though it has demonstrated strong empirical performance in Table 2 and across most additional experiments conducted during rebuttal. Finding an interpolation $U_t = I(t, U_0, U_1)$ better tailored to the underlying problem structure is a promising direction worthy of its own in-depth investigation, as we acknowledged in Sec 6 (L287). Importantly, we emphasize that as long as the interpolation function $I$ is differentiable w.r.t. both $t$ and $x$, our NAAS framework remains fully applicable.
> > > >
> > > > ---
> > > >
> > > > **Closing remarks**
> > > >
> > > > We thank the reviewer again for all valuable comments. As we approach the end of the discussion period (**Aug 8th 11:59pm AOE**), we would like to briefly summarize the author-reviewer discussion.
> > > >
> > > > During the rebuttal period, we have provided extensive additional clarifications and outlined plans to revise certain sections in the manuscript. Additionally, we conducted extensive experiments that provide strong evidence of the robustness and effectiveness of NAAS, including:
> > > >
> > > > - An in-depth ablation study on the mode-weight estimation problem across various dimensions and increasing between-mode distances, demonstrating that NAAS achieves comparable performance to LRDS.
> > > >
> > > > - Integration of AS and LRDS into the benchmarks reported in Table 2, where NAAS consistently outperforms these standard sampling baselines.
> > > >
> > > > - Finally, an ablation study on the exploration hyperparameter $\sigma_t$, where we demonstrate the robustness of NAAS and discuss further improvement by leveraging reference samples, akin to the approach used in LRDS.
> > > >
> > > > We have made every effort to address all questions raised by the reviewer in both the initial review and during the rebuttal. If our responses adequately address your concerns, we kindly ask the reviewer to reconsider their initial rating, so that it better reflects the current stage of the discussion.
> > > >
> > > >
> > > > ---
> > > >
> > > > [9] https://github.com/microsoft/soc-fine-tuning-sd/blob/main/configs/multi_prompt.yaml#L7
> > > > [10] Consistency Models

---

> > > > > ### Comment · Reviewer_GpP4 · 2025-08-09
> > > > > **Final answer**
> > > > >
> > > > > Thank you a lot for providing these additional elements, which legitimate even more the framework of NAAS. I fully agree with the authors on the three elements brought up during this discussion. I suggest to also add: the other elements discussed:
> > > > > - a discussion about the warm up upon prior knowledge (which in practice helps a lot other baselines such as LRDS and AS, and could be beneficial to NAAS)
> > > > > - a discussion about the limitations evoked : use of two neural networks + computation of Hessian of $\log U_t$
> > > > > - a discussion about the extension to a more general density interpolant
> > > > > - a clear comparison of the computational budgets between the methods
> > > > > - a clarification of the motivation of the work, as given by the new introduction (more focused on helping the method by adding a reference rather than avoiding resampling)
> > > > >
> > > > > Thanks to this discussion, I am inclined to increase my score, hoping that all these elements would be integrated in the updated version of the paper.

---

### Official Review · Reviewer_YsLF · 2025-07-04

**Clarity:** 2
**Significance:** 3
**Originality:** 3
**Rating:** 5
**Confidence:** 4

**Summary:**

In this work, the authors present a novel Adjoint Matching approach for sampling from a given unnormalised density. This aligns closely with recent interest in the generative modelling community in moving away from sample-based approaches. Within this framework, the presented theory falls under the category of using a controlled annealed SDE. However, in contrast to most work that uses the annealing interpolation for reweighting, the presented method proposes an interesting approach to provide unbiased samples by considering how to transform the initial distribution.

The proposed method achieves the learning of the controlling force for the SDE, as well as how to sample from the adjusted prior, using two different variants of the Adjoint Matching framework. These two sides of the learning problem are then combined in alternating fashion. A replay buffer is used to reduce simulation overhead.

The method is evaluated on a number of well-established benchmarks, on which it demonstrates excellent performance.

**Questions:**

Please refer to the Strengths and Weaknesses section for a complete discussion of my comments and questions. As stated there: *Pending the authors’ response to my comments on the motivation of the work and the comparisons with reweighting taken into account, I do not feel confident recommending the paper for publication at this time. However, if these points are properly addressed, I would be willing to increase my score to a 5. I will likely not increase my score to a 6, as the work builds heavily on existing variants of adjoint matching and contains limited real-world evaluation. Please see below for a more detailed list of comments and questions.*

**Ethical Concerns:**

["NO or VERY MINOR ethics concerns only"]

**Final Justification:**

In their rebuttal and following discussions the authors clarified a number of points of their paper and with that further established their contribution in my opinion. Based on the initial rebuttal I've increased my score to a 4, and consecutively increased it to a 5 after their final set of comments.

**Limitations:**

As noted by the authors in the conclusion, the method still needs to be tested further on real-world problems. I do not foresee any negative societal impact at this time.

**Paper Formatting Concerns:**

No formatting concerns

**Quality:**

3

**Strengths And Weaknesses:**

The paper presents an interesting approach to an important and currently highly active field of research: sampling from a distribution given its unnormalised density. The paper is generally well written, contains a great deal of technical depth, and demonstrates strong results on a number of well-established benchmarks.

However, the motivation for the proposed approach, and with it the main contribution, is not clearly communicated in the writing. My understanding is the following. Given an SDE with time-dependent density $U_t$, recent work has focused on using importance weighting to debias the drawn samples. This, however, can result in high variance. Adding a control term $u$ can help reduce this variance, but it also requires modifying how the weights are calculated. Specifically, it introduces an additional divergence term and a cross term. These modifications can reduce the variance of the weights, but unless $U_t$ and $u$ together satisfy the continuity equation, the weights will still exhibit some variance. Related work addresses this by directly learning to minimise the controlled weights, and with that the variance of the estimator. In the context of Adjoint Matching, however, minimising the controlled weights is not part of the learning objective. As a result, even after convergence, the weights are not constant. One way to resolve this is to adjust the prior distribution, which is the focus of this paper.

As a side note, and this could be more clearly stressed in the paper, even in the case where $U_t$ and $u_t$ are learned to solve the continuity equation, there remains a $Z_0 / Z_1$ term. The only way to eliminate this term is to adjust the prior. Therefore, changing the prior is not only a way to avoid varying reweighting factors, it is a necessary step to remove the constant weight factor altogether.

If my understanding is correct, I believe this should be explained more clearly in the paper. With this change, the motivation and contributions would become more transparent. The current primary motivation of avoiding reweighting is, in my opinion, not convincing. If calculating the weights is feasible, this always offers an advantage. Given two samplers that provide similar sample quality, a sampler that supports reweighting is always more useful than one that does not.

This also relates to another aspect of the paper that I believe should be reconsidered. In the experiments section, the paper compares several baseline methods, including both those that support reweighting and those that do not. However, the reweighting is not taken into account in any of the evaluations. This is particularly notable for SCLD, the second-best performing method, which does support reweighting. I would be interested to see a comparison on a task where reweighting is beneficial, for example, estimating an expectation.

Pending the authors’ response to my comments on the motivation of the work and the comparisons with reweighting taken into account, I do not feel confident recommending the paper for publication at this time. However, if these points are properly addressed, I would be willing to increase my score to a 5. I will likely not increase my score to a 6, as the work builds heavily on existing variants of adjoint matching and contains limited real-world evaluation. Please see below for a more detailed list of comments and questions.

**Further comments and questions:**

- The paper (and so does prior work) frequently uses the notion of  "non-equilibrium," including in the title. I would appreciate a clearer discussion of what criteria make the method non-equilibrium, and whether these still apply to the proposed approach. Does "non-equilibrium" refer to intermediate instantaneous distributions not matching the distribution defined by $U_t$? The concept of equilibrium becomes more confusing when non-conservative forces such as $u$ are present, as they affect the stationary distribution of the system.

- **Table 1:** The use of check marks and large red crosses, along with the phrasing "reliance," implies a strong negative stance on the use of importance weighting. I do not believe this is justified. Changing this to a simple "Uses IWS" with a yes/no column would, in my opinion, be more balanced.

  **L52:** The statement "ensures unbiasedness by correcting the prior distribution" is surprising here, as there has not yet been any discussion of how adjusting the prior contributes to unbiasedness. A little bit or rewriting could help improve the readability of the introduction significantly here.

- The notation used to index functions by time is not consistent. Most of the time a subscript is used, but there are also cases where time is passed directly to a function. If this was a deliberate choice, it should be clarified.

- **L110–111:** It is not immediately obvious why the value function at initial time becomes constant. A short clarifying remark would help the reader.

- **L156:** The phrase "special interpolation" is unclear, especially since the schedule used is not actually an interpolation.

- **Corollary 3.2:** The claim that DDS is a special case of the proposed method seems questionable. DDS does not use an interpolation and does not alter the prior.

- **Figure 1:** This figure was very helpful. Including a similar illustration in the introduction could greatly improve clarity and reduce early confusion.

- **L207–209:** The discussion of the replay buffer appears to repeat what was said in the previous paragraph.

- **Table 2:** A minor suggestion, but linking each method to its citation would improve the table’s usefulness.

- Regarding the baselines, I would expect NETS to be included given the strong methodological overlap. If it was omitted due to unavailable code, this should be explicitly stated.

- Related, it would be useful for the authors to explain why the MMD scores are not reported for most methods on the Funnel and MoS datasets.

- Many in-text citations do not correctly link to the references at the end of the paper. Clicking them takes the reader to the first or second page instead.

---

> ### Author Rebuttal · Authors · 2025-07-31
>
> We appreciate the reviewer for spending time reading our manuscript carefully and providing valuable feedback. We hope that our responses are helpful in addressing the reviewer’s concerns. Due to the strict character limit, we have grouped and summarized related comments where appropriate.
>
> **1. Clarification on motivation**
>
> *Summary of reviewer’s comments*:
> The reviewer raised the unclear motivation in the current presentation and hypothesized that the motivation may be to propose a diffusion sampler (e.g., NETS [1]) that learns a control $u$ satisfying continuity equation prescribed by the annealing energy $U_t$. Since the proposed method does not minimize variance of weights explicitly (as like LogVar [2]), additional adjustment on prior distribution is necessary. The reviewer also raised questions about reweighting.
>
> **A.** We thank the reviewer for the effort to understand our manuscript. We agree that our motivations need improvement. In this response, we address points that may have been misleading or unclear (1.1-1.3). Based on this clarification, we have revised the corresponding parts of the manuscript to improve transparency and presentation (1.4).
>
> **1.1 The roles of $u$ and $U_t$**
>
> The reviewer understands correctly that, recently, there have been a few annealed-based diffusion samplers that learn a control $u$ to enforce the continuity equation by introducing reweighting terms to align the marginals. A noticeable example is NETS [1]. That is, the role of $u$ for methods like NETS is to learn a control $u$ that matches prescribed marginal $p_t \propto e^{-U_t}$ given annealing energy $U_t$.
>
> However, the aforementioned motivation differs significantly from our proposed method, NAAS. Specifically, the optimal control considered in NAAS needs not match $p_t \propto e^{-U_t}$, except at the terminal time t=1 (see Theorem 3.4). The distinction stems from the fact that, unlike NETS which does not admit Stochastic Optimal Control (SOC) interpretation, NAAS is grounded fundamentally on the SOC framework whose control $u$ is directly optimized to reach the final target distribution, without being constrained to follow any specific path of intermediate marginals. Moreover, thanks to the rich literature on solving SOC problems, we have the flexibility to choose methods, including Adjoint Matching (AM), that do not rely on importance sampling.
>
> **1.2 Discussion on weight-based vs AM-based SOC solvers**
>
> In the case where SOC problems are considered for Boltzmann sampling, there exist prior methods (such as LogVar [2]) that solve the SOC by directly minimizing the variance of the importance weights induced by the controlled dynamics, as mentioned by the reviewer. In contrast, other approaches—including our NAAS—use AM [3,4], which optimizes a matching loss based on the backward adjoint state.
>
> We emphasize that both solvers aim to approximate the same optimal control. Therefore, if both methods are solved optimally, they yield identical solutions, hence, importance weights of optimal solution should be constant. In our work, we adopt AM due to its empirical advantages in variance reduction, scalability, and learning stability.
>
> We also emphasize that employing an AM-based SOC solver does not exclude the use of resampling during inference. We agree with the reviewer that resampling can be particularly beneficial in the evaluation phase, especially when samples produce low-variance estimators, as discussed in Appendix B. While resampling could theoretically be incorporated during NAAS training, in practice, we find that using AM alone is sufficient to achieve strong performance.
>
> **1.3 Why do we need to learn the prior distribution**
>
> The SOC problem in Theorem 3.2 prescribes an initial distribution $\mu$ (see Equation (13)) such that the corresponding terminal distribution matches the target $\nu$. However, $\mu$ is generally intractable. To address this, we introduce an auxiliary SOC problem (Lemma 3.3) to learn a prior distribution that approximates $\mu$. Then, in Theorem 3.4, we show that these two SOC problems can be unified into a single joint problem whose solution produces samples from the target distribution $\nu$ when optimally solved.
>
> **1.4 Clarification of our motivation & contribution**
>
> Rather than positioning our method solely as an annealing-based sampler that avoids importance sampling, we will present it as one of the key characteristics of our approach. Instead, we would like to clarify that the core motivation and contribution of our work lies in developing a SOC-based sampler that incorporates annealed reference dynamics, as illustrated in the following flow:
>
> - Prior SOC-based approaches [4,7,8] have constructed unbiased diffusion samplers using stationary or uninformative base references, and do not fully leverage annealing references as employed in other importance sampling based methods. In this work, we propose a novel SOC-based framework that employs annealed reference dynamics as the base SDE.
> - (*) Precisely, we formulate two SOC subproblems: (i) learning a prior distribution $\mu$ from the initial distribution $\delta_0$ with standard base reference, and (ii) transporting $\mu$ to the target $\nu$ using controlled annealed dynamics. We unify these into a single SOC formulation, which yields a controlled diffusion process that effectively transports $\delta_0 \to \mu \to \nu$.
> - Among available solvers for SOC, we adopt AM [3], a recently proposed method being scalable and sample-efficient. As discussed in Adjoint Sampling [4], this approach enables scalability and improved training stability.
>
> **2. Clarification on reweighting**
>
> **A**. Thank you for raising this important point. We realize that our manuscript may have caused some confusion regarding the role of reweighting. Specifically, when we stated that a method "relies on importance sampling," we were referring to the use of importance sampling during training. Our key claim is that, our method does not require importance sampling during training. This distinction is important because using importance sampling during training often leads to high-variance [5,6], which can negatively affect performance. We will revise the manuscript to clarify this distinction in the abstract, introduction, and Table 1.
>
> Regarding your point about evaluation: we appreciate the suggestion to consider reweighting at test time. Like other SOC-based samplers [4,7], our method does support reweighting during evaluation, as discussed in Appendix B. However, in this work, we chose not to emphasize evaluation-time reweighting to focus on the properties of the learned samplers themselves. We agree that incorporating reweighting into evaluation could be informative and plan to explore this direction in future work.
>
> **3. Clarification on remaining questions**
>
> *Q1*. The meaning of “non-equilibrium”.
>
> *A.* In equilibrium-based samplers such as Langevin dynamics, the stochastic process is constructed so that its stationary distribution coincides with the target distribution. Sampling is achieved by simulating the process over a sufficiently long (often asymptotic) time horizon. These dynamics eventually “equilibrate” to the target, and accurate sampling is guaranteed only in the infinite-time limit. In contrast, non-equilibrium samplers explicitly construct a finite-time dynamical system whose terminal distribution matches the target, without requiring convergence to a stationary state. In our method, NAAS, we formulate the sampling task as a SOC problem, where the control is optimized to ensure that the terminal distribution at time $t=1$ matches the desired target distribution $\nu$.
>
> *Q2*. In Table 1, phrasing "reliance," implies a strong negative stance. Changing this to a simple "Uses IWS" with a yes/no column would be more balanced.
>
> *A*. We agree with the reviewer’s comment. We revised the table to use the more neutral label “Uses IWS during training”.
>
> *Q3*. L52: "ensures unbiasedness by correcting the prior distribution" is surprising here, as there has not yet been any discussion of how adjusting the prior contributes to unbiasedness. A little bit or rewriting could help improve the readability.
>
> *A*. We agree that introducing the notion of "unbiasedness" at that point in the introduction was insufficiently motivated. To enhance clarity and improve the logical flow, we revised the corresponding paragraph, now presented as paragraph (*) in the “1.4 Clarification of Our Motivation & Contribution” section.
>
> *Q4*. The notation used to index functions by time is not consistent.
>
> *A*. As the reviewer suggested, we put every time variable into the subscript. For example, we fixed time variables in adjoint terms and neural networks as subscripts.
>
> *Q5*. Typo / Minor comments on L110–111, L156, Corollary 3.2, Figure 1, Table 2, citation/reference.
>
> *A*. We thank the reviewer for carefully reading our paper and providing valuable suggestions. We will reflect all these suggestions to clarify our manuscript. Due to the character constraints of this rebuttal, we are unable to detail all the revisions here. However, we would like to highlight that most of the metric values in Table 2 were taken from the existing papers, and MMD scores are not reported for many methods in their original works. We will include this explanation in our manuscript to ensure clarity. Additionally, we will correct the link to the reference as suggested.
>
> Once again, thank you for your careful review and thoughtful suggestions. We hope that the revisions we’ve made address your concerns.
>
> ---
> [1] Non-equilibrium Transport Sampler
>
> [2] Solving high-dim HJB PDEs using neural networks
>
> [3] Adjoint Matching
>
> [4] Adjoint Sampling
>
> [5] Sequential Monte Carlo Samplers
>
> [6] Free energy computations: A mathematical perspective
>
> [7] Path Integral Sampler: a stochastic control approach for sampling
>
> [8] Denoising Diffusion Samplers
>
> [9] Sequential Controlled Langevin Diffusions

---

> > ### Comment · Reviewer_YsLF · 2025-08-06
> > **Increase score, but still some issues**
> >
> > I thank the authors for their response. Based on the rebuttal I will increase my score to a 4. My primary reason for increasing my score is that I believe that by learning the reference distribution the authors address a very interesting and important problem, but I still remain of the believe that this is not sufficiently motivated in the paper when taking into account the promised changes.
> >
> > Furthermore, there are still a few topics that I believe need to be clarified further before I can increase my score to a 5 as mentioned in the original review.
> >
> > Most notably, I am not satisfied with the comments regarding not needing a pre-specified reference distribution and corresponding energy $U_t$. This is a crucial component of the reference path that the authors aim to match (11).
> >
> > Second, I do not agree with the authors definition for Non-Equilibrium sampling. In my opinion, non-equilibrium methods are those that do not require the final instantaneous distribution to match the desired target distribution (as is the case in this work), but still allows to make meaningful inference despite this. AIS is the clearest example of this in ML, but it can also be seen in the original physics context. Jarzynski's equality as well as Crooks fluctuation theorem are clear examples from Non-Equilibrium Thermodynamics where the samples at time $t$ never match the equilibrium distribution, but yet we can make interesting claims about the difference in the free-energy differences.

---

> > > ### Author Response · Authors · 2025-08-08
> > >
> > > We greatly appreciate reviewer’s feedback and the raise of score. We are excited that the reviewer has recognized the importance of the problem formulated by NAAS. We provide additional clarifications below.
> > >
> > > ---
> > >
> > > **1. Pre-specified reference paths (Eq 11)**
> > >
> > > - As the reviewer mentioned, the pre-specified, annealing reference path in Eq 11 is indeed the key component to our method, and we fully agree with the reviewer on this assessment. That said, it was _never_ our intention to omit or downplay the importance of reference path (Eq 11). Rather, the motivation of our initial rebuttal—acknowledging that it might have been unclearly written—was to provide additional clarification between our NAAS and prior diffusion samplers—particularly NETS [1]—that are based on the same reference path (Eq 11).
> > >
> > >
> > > - To clarify, both NETS and NAAS are based on the _same_ reference path (Eq 11). Consequently, these pre-specified reference paths play a crucial role in both methods. Since the instantaneous (i.e., marginal) distribution induced by Eq 11 at any time $t \in (0,1]$ need _not_ match the reference distribution $\exp(-U_t)$, in NETS, an additional term $u$ is introduced to the original drift in Eq 11 to eliminate the discrepancy. That is, the $u$ proposed by NETS modifies the reference path (Eq 11) such that the resulting path yields the reference distribution $\exp(-U_t)$ at all time $t \in [0,1]$.
> > >
> > >
> > > - On the other hand, while NAAS also appends an additional term $u$ to the same reference path (Eq 11), its $u$ is derived within the stochastic optimal control framework, aiming to solve a specific objective formulated in Eq 12. This distinction alters the effect of $u$ in modifying the reference path (Eq 11) between the two methods, as the objective of NAAS (Eq 12) does not explicitly enforce $u$ to match the reference distribution $\exp(-U_t)$. Indeed, we prove in Sec 3.1 (between L146 and 147) that the $u$ proposed by NAAS results in a controlled SDE whose instantaneous distribution $p_t$ satisfies $p_t \propto \exp(-U_t - V_t)$, where $V_{t=1} = 0$ (see L135). That is, the instantaneous distribution of NAAS—as expected by the reviewer—is _highly correlated_ with the reference distribution $\exp(-U_t)$, but is _biased_ by an additional term $\exp(- V_t)$, which vanishes at the terminal time $t=1$.
> > >
> > >
> > > - We highlight that since $V_{t=1} = 0$, it follows that $p_1 \propto \exp(-U_1 - V_1) = \exp(-U_1)$. Hence, the controlled SDEs proposed by NAAS do converge to the desired target distribution $\nu \propto \exp(-U_1)$ at terminal time $t=1$. Permitting an additional bias $V_t$ for $t < 1$ enables the formulation of $u$ within the stochastic optimal control framework. In return, it facilitates application of adjoint matching [3], which has proven effective for Boltzmann distribution sampling in recent work like AS [4], as well as our proposed NAAS. We will include these clarifications in the revision, and we thank the reviewer for raising the questions.
> > >
> > > ---
> > >
> > > **2. Non-equilibrium sampling**
> > >
> > > - We thank the reviewer for the comments. We fully agree with the reviewer that the “non-equilibrium” methods should _not_ be simply characterized as those that “do not aim for equilibrate distribution at intermediate time $t$”. As the reviewer points out, such a naive definition overlooks the subtle concepts behind. Instead, we acknowledge that, in the similar spirit of AIS and thermodynamics, “non-equilibrium” samplers are more appropriately described as methods that progressively guide samples over a finite horizon using a meaningful reference annealing path defined by the annealing energy $U_t$ between target and prior distributions—for instance, the reference path (Eq 11) considered by NAAS.
> > >
> > > - Such guidance from non-equilibrium—or reference—distribution $\exp(-U_t)$ indeed plays a key role in NAAS. Specifically, as we discussed in the third and fourth bullet points above, the instantaneous (i.e., marginal) distribution $p_t \propto \exp(-U_t - V_t)$ of NAAS is strongly affected by the reference distribution $\exp(-U_t)$, with the two distributions coinciding exactly at the terminal time $t=1$, i.e., $p_1 \propto \exp(-U_1)$. We will include these discussions in the revision, and we are grateful to the reviewer for helping to enhance the clarity of our presentation.

---

> > > > ### Comment · Reviewer_YsLF · 2025-08-08
> > > > **Clear comments, will increase score further**
> > > >
> > > > I thank the authors for the final set of clarifying comments, both of which help me significantly better understand the paper.
> > > >
> > > > If the authors would be able to include a (shortened) version of both these comments in the paper, I believe that this will make it significantly easier to follow. With these changes in mind, I will increase my score further to a 5.

---

### Official Review · Reviewer_sTeb · 2025-07-06

**Clarity:** 2
**Significance:** 2
**Originality:** 2
**Rating:** 3
**Confidence:** 3

**Summary:**

his paper presents a Non-Equilibrium Annealed Adjoint Sampler, a stochastic optimal control-based diffusion method that enables scalable training without resorting to importance sampling. To this end, the authors make the following contributions:
1. The authors introduce a stochastic optimal control-based unbiased sampler where the reference dynamics is governed by an annealed stochastic differential equation.

2. The authors develop an adjoint matching-based optimization algorithm that updates two control functions in alternation, enabling scalable training while avoiding importance sampling and its associated variance issues.

**Questions:**

1. Could you provide scalability experiments on synthetic problems of increasing dimension, comparing, for example, wall-clock time or memory usage against importance sampling and other baselines?

2. Could you describe whether a joint optimization approach for both control functions is feasible, and if so, provide empirical evidence comparing the alternating scheme against joint optimization? Additionally, could you include a discussion or empirical evidence about  convergence diagnostics and discuss potential failure modes of the alternating optimization scheme?

3. Could you provide empirical variance analysis comparing your method's variance reduction against, for example, importance sampling-based methods?

4. Could you include the missing reference for the SMC baseline used in your experiments and ensure all baselines are properly cited?

5. The paper would benefit from a presentation review to improve clarity.

6. Could you discuss the performance (in particular, generalization) of the proposed method on perhaps more diverse molecular systems beyond the alanine dipeptide?

**Ethical Concerns:**

["NO or VERY MINOR ethics concerns only"]

**Final Justification:**

The authors have provided a good rebuttal. The responses provide new and valuable experimental data, and agree to make several important clarifications in the manuscript. In particular, the new scalability experiments strengthen their paper. However, the authors are somewhat evasive on key requests for direct empirical comparisons (alternating vs. joint optimization, direct variance measurements), offering theoretical justifications or re-interpreting existing results. While their arguments are plausible, the lack of direct evidence for these specific points remains a minor weakness.

**Limitations:**

Yes

**Paper Formatting Concerns:**

No formatting concerns.

**Quality:**

3

**Strengths And Weaknesses:**

Strengths:
1. The paper presents effectively the stochastic optimal control formulation and the adjoint matching methodology.

2. The paper provides strong theoretical results: Theorem 3.1 demonstrates that the SOC-based approach yields the desired sampler, while Theorem 3.4 establishes the crucial unbiasedness property of the sampler.

3. The authors choose appropriate test problems: synthetic multi-modal distributions for rigorous quantitative evaluation and alanine dipeptide generation for practical molecular applications.

4. In the appendix, the authors study the effect of annealing schedule on training performance.

Weakness:
1. Having two learnable control functions effectively doubles the parameter count, which may limit scalability to high-dimensional problems.

2. While the replay buffer may reduce computational complexity, the implementation of the proposed Algorithm 1 appears complex and could introduce stability issues. The alternating optimization scheme may face convergence challenges that are not discussed.

3. Despite claiming scalability as a key contribution, the paper lacks empirical scalability analysis. No experiments test performance on high-dimensional problems.

4. The paper does not discuss potential failure modes of the alternating update scheme or provide empirical evidence comparing scalability against importance sampling baselines. Additionally, no variance reduction measurements are provided, despite this being a claim in the abstract over importance sampling methods.

---

> ### Author Rebuttal · Authors · 2025-07-31
>
> We appreciate the reviewer for spending time reading our manuscript carefully and providing valuable feedback. We hope that our responses are helpful in addressing the reviewer’s concerns. Due to the strict character limit, we have grouped related comments and questions where appropriate.
>
> $ $
>
> ---
> **1. Clarification on alternating vs joint optimization (W1, W2, Q2)**
>
> *Summary of reviewer’s comments*:
> The reviewer raised concerns about scalability due to the doubled parameter count from two learnable control functions (W1) and the complexity and stability of Algorithm 1 (W2). They suggest considering a joint optimization approach and discussing convergence diagnostics and potential failure modes (Q2).
>
> *Author response*:
> We thank the reviewer for these insightful questions. We agree that the choice between joint vs alternating optimization is a central design decision. Here, we provide further clarification on motivation of alternate optimization.
> The optimization of NAAS occurs in two stages. The first stage, over $t$=[−1,0], learns a broad, smooth prior distribution $\mu= e^{- U_0 - V_0}$ (where $U_0$ is typically the energy of Gaussian) that broadly covers the target distribution $\nu$. That is, rather than capturing all fine-scale structures, the first stage aims to construct a low-complexity distribution with broad support over $\nu$. The second stage, over [0, 1], then refines the dynamics to guide trajectories from $\mu$ to $\nu$. As annealed dynamics already bias toward high-density regions, this stage learns to correct residual mismatches and is easy to learn.
>
> Decoupling these objectives improves stability and training. Each stage solves a simpler, focused subproblem: the first creates a good initialization, and the second incrementally corrects the flow. This separation reduces interference, enhancing convergence and sample quality.
>
> Finally, regarding the reviewer’s concern about implementation complexity, we would like to clarify that our algorithmic structure is straightforward. For both 1st and 2nd stages, we (i) roll out trajectories, (ii) compute adjoint variables via standard backward integration, and (iii) update the networks using buffered samples with the stable adjoint-matching loss. These procedures are modular and computationally efficient.
>
> We have incorporated a discussion of these design choices, results, and interpretation in the Appendix and have also referenced it in the main text for clarity.
>
> $ $
>
> ---
> **2. Additional experiments on higher-dim problems (W3, W4, Q1)**
>
> *Summary of reviewer’s comments*:
> The reviewer noted a lack of empirical scalability analysis (W3), requested comparison on higher-dim problems with other baselines (Q1), and potential failure modes (W4).
>
> *Author response*:
> We thank the reviewer for highlighting the importance of evaluating scalability and for suggesting a more detailed empirical analysis on high-dimensional settings.
>
> While we agree that scalability is a crucial aspect, we kindly note that our current experiments readily include challenging, high-dimensional problems in sampling literature [1-4] where ground-truth samples are obtainable. Specifically, the GMM40 and MoS datasets in Table 2 are 50 dim, whereas the alanine dipeptide in Table 3 has 60 dim. In all settings, our NAAS consistently outperforms existing baselines by a substantial margin, achieving over 2× improvement in sample quality over the second-best method. Importantly, we compare against a wide range of importance sampling–based baselines, including SMC, SMC-ESS, and SCLD [4]. Our method not only achieves better average performance and more robust convergence across random seeds.
>
> In response to the reviewer’s suggestion, we conduct additional experiments on 100-dim GMM40 tasks. Specifically, we evaluated our method and two strong baselines: Adjoint Sampling [9], which uses standard reference dynamics, and SCLD [4], a SOTA importance sampling method. We also report wall-clock time and peak memory usage for 50-dim experiment, measured on 1 A6000 GPU. The results are summarized below:
>
> | Method | $d=4$ | $d=16$ | $d=50$ | $d=100$ | Mem. usage. | run time |
> |----|----|----|----|----|----|----|
> | SCLD | 157.90 | 1033.19 | 3787.73 | 12357.49 | - | 30-40 mins |
> | AS | -  | - | 18984.21 | - | 1GB | 30 mins |
> | Ours | **25.15** | **37.96** | **496.48**  | **633.20** | 1GB | 120 mins |
>
> Our method remains robust and maintains strong performance even as dimensionality increases, while the baseline, SCLD, experience significant degradation as the dimensionality grows. Furthermore, our method demonstrates competitive runtime and memory consumption, remaining scalable as dimensionality increases. This is partly due to the use of replay buffers, which effectively ensures efficient training. We acknowledge, however, that a limitation of our method is the need to solve a backward adjoint state (Equation 22), which contributes to the longer training time. Nonetheless, we believe that the resulting performance gains, especially in sample quality, more than justify this additional computational cost.
>
> Regarding potential failure modes of the alternating scheme, we would like to note that across all tasks, including the high-dimensional ones, we did not observe signs of divergence or instability when properly tuned. We attribute this to the structured separation of roles in the alternating updates, which helps stabilize training dynamics, as discussed in our response to Q2.
>
> $ $
>
> ---
> **3. Discussion on variance reduction (W4, Q3)**
>
> *Author response*:
> We thank the reviewer for raising this important point. We recognize that the phrasing in the abstract and introduction “reliance on importance sampling leads to high variance” may have misled readers. Hence, we would like to clarify on the high-variance property in importance sampling (IS)–based algorithms and why our method is less susceptible to this issue.
>
> Formally, IS methods approximate the target distribution by $\nu\approx\sum^N_{i=1} w^{(i)}\delta_{X^{(i)}}$, where the unnormalized weights are obtained by $w^{(i)} \propto \frac{dp^\star}{dp} (X^{(i)})$ with the ratio between the target path measure $p^\star$ and proposal $p$. When the support or mass of the proposal and target distributions are misaligned—as is common in high-dimensional or complex energy landscapes—these weights can suffer from high variance, as noted in prior works (e.g. [5]; [6] Section 4.1.4). This leads to poor effective sample size and significant variance in the empirical approximation, and can be further exacerbated when the number of samples $N$ is insufficient.
>
> For example, in PDDS [3], the learned distribution approximates a reweighted empirical measure and is thus exposed to the instability induced by high-variance weights. In contrast, our method learns a sampling distribution directly via path-space control, thereby avoiding the need to compute or rely on importance weights.
>
> Additionally, as shown in Table 2, SMC-based methods (including SMC, CRAFT [7], and SCLD [4]) exhibit substantially higher variance on the 50-dimensional GMM40 and MoS benchmarks, especially in comparison to PIS [8], AS [9], and our method, which do not rely on importance weighting. We will clarify this point in the revised manuscript and provide additional discussion to explicitly contrast variance behaviors.
>
> $ $
>
> ---
> **4. Clarification on remaining questions**
>
> **Q4**: *Missing SMC references*
>
> *Author response*: We thank the reviewer for pointing this out. We follow the SMC and SMC-ESS implementation in [4]. We will incorporate the explanation in Line 238.
>
> **Q5**: *Presentation*
>
> *Author response*: We thank the reviewer for their detailed comments, including the helpful suggestion regarding presentation clarity. Following the reviewer’s feedback, we have revised the manuscript to improve overall clarity and structure. In particular, we have
> clarified the stability and scalability aspects of our method,
> Expanded the discussion on design choices (e.g., alternating vs. joint optimization), and
> Incorporated additional discussion on importance sampling and its variance
> as raised in W1-5. We again thank the reviewer for helping us clarify and improve the manuscript. We believe these revisions significantly enhance the clarity and accessibility of our work, and better highlights our key contributions.
>
> **Q6**: *Generalization of NAAS beyond alanine dipeptide (ALDP)*
> *Author response*: We agree with the reviewer that evaluating the generalization of our method on more diverse and larger-scale molecular systems beyond ALDP would further strengthen the paper. Due to the limited time available during the rebuttal period and the significant computational cost associated with high-dimensional molecular simulations, we were not able to include such additional experiments. However, we acknowledge this as a limitation and have added a corresponding discussion in the revised manuscript.
>
> On the other hand, we would like to note that ALDP remains a widely used and nontrivial benchmark in the molecular sampling literature. With approximately 60 dimensions, it already presents meaningful challenges.
>
> ---
> **References**
>
> [1] Improved Sampling via Learned Diffusions
>
> [2] Non-equilibrium Transport Sampler
>
> [3] Particle Denoising Diffusion Sampler
>
> [4] Sequential Controlled Langevin Diffusions
>
> [5] Sequential Monte Carlo Samplers
>
> [6] Free Energy Computations: A Mathematical Perspective
>
> [7] Continual Repeated Annealed Flow Transport Monte Carlo
>
> [8] Path Integral Sampler
>
> [9] Adjoint Sampling

---

> ### Comment · Reviewer_sTeb · 2025-08-06
> **While their arguments are plausible, the lack of direct evidence for these specific points remains a minor weakness**
>
> The authors have provided a good rebuttal. The responses provide new and valuable experimental data, and agree to make several important clarifications in the manuscript. In particular, the new scalability experiments strengthen their paper. However, the authors are somewhat evasive on key requests for direct empirical comparisons (alternating vs. joint optimization, direct variance measurements), offering theoretical justifications or re-interpreting existing results. While their arguments are plausible, the lack of direct evidence for these specific points remains a minor weakness.

---

> > ### Author Response · Authors · 2025-08-08
> >
> > We thank the reviewer again for all valuable comments. We are excited that the reviewer has found our responses adequately addressed all the questions raised in the initial reviews. As we approach the end of the discussion period (**Aug 8th 11:59pm AOE**), we would like to ask whether the reviewer requires any additional clarifications and/or discussion.
> >
> > If our replies adequately address your concerns, we kindly ask the reviewer to reconsider their initial rating, so that it better reflects the discussion at the current stage.

---

> > ### Author Response · Authors · 2025-08-08
> >
> > We are grateful to the reviewer for acknowledging our additional experiments and clarifications. Below, we provide two additional experiments suggested by the reviewer.
> >
> > - **Additional experiments on alternating vs joint optimization.** The table below compares the alternating optimization scheme and the joint optimization scheme—i.e., optimizing the entire horizon $t \in [-1, 1]$ simultaneously—on two benchmarks. Notably, our proposed alternating optimization scheme consistently outperforms the joint optimization. This empirical finding aligns well with our theoretical conjecture in the initial rebuttal. We will include these results in the revised ablation study section to provide a comprehensive evaluation. We thank the reviewer for raising this topic.
> >
> >   | Sinkhorn ($\downarrow$) | MW54 | MoS |
> >   |----|----|----|
> >   | Joint | 0.20 | 597.99 |
> >   | Alternating (ours) | 0.10 |394.55  |
> >
> >
> > - **Additional experiments on variance measurement.** In the table below, we report the variance of the importance weights across various benchmarks. We choose LV-PIS as the baseline for comparison as the method directly optimizes the variance of importance weights and, similar to our NAAS, starts the generation from Dirac delta. For NAAS, we compute its importance weights using Eq 59 (see Appendix B). The table below shows that the variance of the importance weights for NAAS is significantly smaller than that of LV-PIS. Remarkably, NAAS achieves this reduction in variance despite never explicitly optimizing the variance of importance weights. This empirical finding again aligns well with our theoretical conjecture in the initial rebuttal, and we will include them in the revision.
> >
> >   | Variance of importance weights ($\downarrow$) | MW54 | Funnel |
> >   |----|----|----|
> >   | LV-PIS | 21.11 | 5.46 |
> >   | NAAS (ours) | 1.84 |  0.11 |
> >
> >
> > We hope that our additional results have sufficiently addressed the reviewer’s questions. We always appreciate the reviewer’s active participation in the author-reviewer discussion, which has helped us further improve both the clarity and rigor of our paper.

---

### Note · Authors · 2025-08-13

We thank all PCs, senior ACs, ACs and reviewers for their efforts throughout the review process.

Our submission proposes a new **diffusion model for sampling Boltzmann distribution**. It yields **strong performance** over prior methods and is rigorously grounded in **stochastic control** theory. We believe our method takes a significant step toward **adjoint-based** sampling methods enhanced with **annealing reference dynamics**.

Our rebuttal discussions with the reviewers have been a very fruitful process, and we have committed to include all additional clarifications—including those suggested by the reviewers—into the revision. Furthermore, we conducted **extensive additional experiments** that provide strong evidence of the robustness and effectiveness of NAAS, including:

- Ablation study on scalability, where NAAS outperforms SCLD and AS as dimensionality increases, with comparable runtime and memory complexity.

- Ablation study comparing our proposed alternating scheme over joint optimization, where the alternating scheme consistently yields stronger performance.

- Ablation study on the variance of importance weights, demonstrating that NAAS achieves significantly reduced variance compared to LV-PIS.

- Ablation study on the mode-weight estimation problem across various dimensions and increasing between-mode distances, demonstrating that NAAS achieves comparable performance to LRDS.

- Ablation study on the exploration hyperparameter $\sigma_t$, where we demonstrate the robustness of NAAS.

- Integration of AS and LRDS into the benchmarks reported in Table 2, as well as PIS into ALDP benchmark, where NAAS consistently shows strong performance.

- Ablation study on a variant of NAAS without annealing paths, whose degraded performance highlights the critical role of annealing paths in NAAS.


We have made every effort to address all questions raised by the reviewers during both the initial review and the rebuttal period. We appreciate all comments and suggestions that have helped improve the rigor, clarity, and empirical soundness of our submission.


Thanks again for your consideration.

Best,
Authors of Submission 23219

---

### Decision · Program_Chairs · 2025-09-17

**Decision:**

Accept (poster)

**Comment:**

The paper introduces a stochastic optimal control-based diffusion method. It is based on two stages where the first stage learns a distribution easy to sample from a simple base, and the second refines it towards the target distribution. It shows improved performance against baselines on standard benchmarks, showing reduced variance with respect to importance sampling. The method is however based on previous work on the Adjoint Matching procedure, and is expensive to train since it requires to learn two networks and to solve a backward adjoint state.
The rebuttal introduced important clarifications on motivation and positioning of the work, as well as additional experiments, with all reviewers agreeing on the contributions made by this paper.